# MULTI-AGENT SEQUENTIAL DECISION-MAKING VIA COMMUNICATION

## ABSTRACT

Communication helps agents to obtain information about others so that better coordinated behavior can be learned. Some existing work communicates predicted future trajectory with others, hoping to get clues about what others would do for better coordination. However, circular dependencies sometimes can occur when agents are treated synchronously so it is hard to coordinate decision-making. In this paper, we propose a novel communication scheme, *Sequential Communication* (SeqComm). SeqComm treats agents asynchronously (the upper-level agents make decisions before the lower-level ones) and has two communication phases. In negotiation phase, agents determine the priority of decision-making by communicating hidden states of observations and comparing the value of intention, which is obtained by modeling the environment dynamics. In launching phase, the upper-level agents take the lead in making decisions and communicate their actions with the lower-level agents. Theoretically, we prove the policies learned by SeqComm are guaranteed to improve monotonically and converge. Empirically, we show that SeqComm outperforms existing methods in various multi-agent cooperative tasks.

## 1 INTRODUCTION

The partial observability and stochasticity inherent to the nature of multi-agent systems can easily impede the cooperation among agents and lead to catastrophic miscoordination (Ding et al., 2020). Communication has been exploited to help agents obtain extra information during both training and execution to mitigate such problems (Foerster et al., 2016; Sukhbaatar et al., 2016; Peng et al., 2017). Specifically, agents can share their information with others via a trainable communication channel.

Centralized training with decentralized execution (CTDE) is a popular learning paradigm in cooperative multi-agent reinforcement learning (MARL). Although the centralized value function can be learned to evaluate the joint policy of agents, the decentralized policies of agents are essentially independent. Therefore, a coordination problem arises. That is, agents may make sub-optimal actions by mistakenly assuming others' actions when there exist multiple optimal joint actions (Busoniu et al., 2008). Communication allows agents to obtain information about others to avoid miscoordination. However, most existing work only focuses on communicating messages, *e.g.,* the information of agents' current observation or historical trajectory (Jiang & Lu, 2018; Singh et al., 2019; Das et al., 2019; Ding et al., 2020). It is impossible for an agent to acquire other's actions before making decisions since the game model is usually synchronous, *i.e.*, agents make decisions and execute actions simultaneously. Recently, intention or imagination, depicted by a combination of predicted actions and observations of many future steps, has been proposed as part of messages (Kim et al., 2021; Pretorius et al., 2021). However, circular dependencies can still occur, so it may be hard to coordinate decision-making under synchronous settings.

A general approach to solving the coordination problem is to make sure that ties between equally good actions are broken by all agents. One simple mechanism for doing so is to know exactly what others will do and adjust the behavior accordingly under a unique ordering of agents and actions (Busoniu et al., 2008). Inspired by this, we reconsider the cooperative game from an asynchronous perspective. In other words, each agent is assigned a priority (*i.e.,* order) of decision-making each step in both training and execution, thus the Stackelberg equilibrium (SE) (Von Stackelberg, 2010) is naturally set up as the learning objective. Specifically, the upper-level agents make decisions before the lower-level agents. Therefore, the lower-level agents can acquire the actual actions of the upper-level agents by

communication and make their decisions conditioned on what the upper-level agents would do. Under this setting, the SE is likely to be Pareto superior to the average Nash equilibrium (NE) in games that require a high cooperation level (Zhang et al., 2020). However, *is it necessary to decide a specific priority of decision-making for each agent?* Ideally, the optimal joint policy can be decomposed by any orders (Wen et al., 2019), *e.g.,* $\pi^*(a_1, a_2|s) = \pi^*(a_1|s)\pi^*(a_2|s, a_1) = \pi^*(a_2|s)\pi^*(a_1|s, a_2)$. But during the learning process, it is unlikely for agents to use the optimal actions of other agents for gradient calculation, making it still vulnerable to the relative overgeneralization problem (Wei et al., 2018). Overall, there is no guarantee that the above equation will hold in the learning process, thus ordering should be carefully concerned.

In this paper, we propose a novel model-based multi-round communication scheme for cooperative MARL, *Sequential Communication* (SeqComm), to enable agents to explicitly coordinate with each other. Specifically, SeqComm has two-phase communication, negotiation phase and launching phase. In the negotiation phase, agents communicate their hidden states of observations with others simultaneously. Then they are able to generate multiple predicted trajectories, called *intention*, by modeling the environmental dynamics and other agents' actions. In addition, the priority of decision-making is determined by communicating and comparing the corresponding values of agents' intentions. The value of each intention represents the rewards obtained by letting that agent take the upper-level position of the order sequence. The sequence of others follows the same procedure as aforementioned with the upper-level agents fixed. In the launching phase, the upper-level agents take the lead in decision-making and communicate their actual actions with the lower-level agents. Note that the actual actions will be executed simultaneously in the environment without any changes.

SeqComm is currently built on MAPPO (Yu et al., 2021). Theoretically, we prove the policies learned by SeqComm are guaranteed to improve monotonically and converge. Empirically, we evaluate SeqComm on a set of tasks in multi-agent particle environment (MPE) (Lowe et al., 2017) and StarCraft multi-agent challenge (SMAC) (Samvelyan et al., 2019). In all these tasks, we demonstrate that SeqComm outperforms prior communication-free and communication-based methods. By ablation studies, we confirm that treating agents asynchronously is a more effective way to promote coordination and SeqComm can provide the proper priority of decision-making for agents to develop better coordination.

## 2 RELATED WORK

**Communication.** Existing studies (Jiang & Lu, 2018; Kim et al., 2019; Singh et al., 2019; Das et al., 2019; Zhang et al., 2019; Jiang et al., 2020; Ding et al., 2020; Konan et al., 2022) in this realm mainly focus on how to extract valuable messages. ATOC (Jiang & Lu, 2018) and IC3Net (Singh et al., 2019) utilize gate mechanisms to decide when to communicate with other agents. Many works (Das et al., 2019; Konan et al., 2022) employ multi-round communication to fully reason the intentions of others and establish complex collaboration strategies. Social influence (Jaques et al., 2019) uses communication to influence the behaviors of others. I2C (Ding et al., 2020) only communicates with agents that are relevant and influential which are determined by causal inference. However, all these methods focus on how to exploit valuable information from current or past partial observations effectively and properly. More recently, some studies (Kim et al., 2021; Du et al., 2021; Pretorius et al., 2021) begin to answer the question: can we favor cooperation beyond sharing partial observation? They allow agents to imagine their future states with a world model and communicate those with others. IS (Pretorius et al., 2021), as the representation of this line of research, enables each agent to share its intention with other agents in the form of the encoded imagined trajectory and use the attention module to figure out the importance of the received intention. However, two concerns arise. On one hand, circular dependencies can lead to inaccurate predicted future trajectories as long as the multi-agent system treats agents synchronously. On the other hand, MARL struggles in extracting useful information from numerous messages, not to mention more complex and dubious messages, *i.e.,* predicted future trajectories.

Unlike these works, we treat the agents from an asynchronously perspective therefore circular dependencies can be naturally resolved. Furthermore, agents only send actions to lower-level agents besides partial observations to make sure the messages are compact as well as informative.

**Coordination.** The agents are essentially independent decision makers in execution and may break ties between equally good actions randomly. Thus, in the absence of additional mechanisms, different

agents may break ties in different ways, and the resulting joint actions may be suboptimal. Coordination graphs (Guestrin et al., 2002; Böhmer et al., 2020; Wang et al., 2021b) simplify the coordination when the global Q-function can be additively decomposed into local Q-functions that only depend on the actions of a subset of agents. Typically, a coordination graph expresses a higher-order value decomposition among agents. This improves the representational capacity to distinguish other agents' effects on local utility functions, which addresses the miscoordination problems caused by partial observability. Another general approach to solving the coordination problem is to make sure that ties are broken by all agents in the same way, requiring that random action choices are somehow coordinated or negotiated. Social conventions (Boutilier, 1996) or role assignments (Prasad et al., 1998) encode prior preferences towards certain joint actions and help break ties during action selection. Communication (Fischer et al., 2004; Vlassis, 2007) can be used to negotiate action choices, either alone or in combination with the aforementioned techniques. Our method follows this line of research by utilizing the ordering of agents and actions to break the ties, other than the enhanced representational capacity of the local value function.

## 3 PROBLEM FORMULATION

**Cost-Free Communication.** The decentralized partially observable Markov decision process (Dec-POMDP) can be extended to explicitly incorporate broadcasting observations. The resulting model is called multi-agent POMDP (Oliehoek et al., 2016).

Pynadath & Tambe (2002) showed that under cost-free communication, a joint communication policy that shares local observations at each stage is optimal. Many studies have also investigated sharing local observations in models that are similar to multi-agent POMDP (Pynadath & Tambe, 2002; Ooi & Wornell, 1996; Nair et al., 2004; Roth et al., 2005a;b; Spaan et al., 2006; Oliehoek et al., 2007; Becker et al., 2004). These works focus on issues other than communication cost and we foucs on the coordination problem. Note that even under multi-agent POMDP where agents can get joint observations, coordination problem can still arise (Busoniu et al., 2008). Suppose the centralized critic has learnt actions pairs $[a_1, a_2]$ and $[b_1, b_2]$ are equally optimal. Without any prior information, the individual policies $\pi_1$ and $\pi_2$ learnt from the centralized critic can break the ties randomly and may choose $a_1$ and $b_2$, respectively.

**Multi-Agent Sequential Decision-Making.** We consider fully cooperative multi-agent tasks that are modeled as multi-agent POMDP, where $n$ agents interact with the environment according to the following procedure, which we refer to as *multi-agent sequential decision-making*.

At each timestep $t$, assume the priority (*i.e.,* order) of decision-making for all agents is given and each priority level has only one agent (*i.e.,* agents make decisions one by one). Note that the smaller the level index, the higher priority of decision-making is. The agent at each level $k$ gets its own observation $o_t^k$ drawn from the state $s_t$, and receives messages $\boldsymbol{m}_t^{-k}$ from all other agents, where $\boldsymbol{m}_t^{-k} \triangleq \{\{o_t^1, a_t^1\}, \dots, \{o_t^{k-1}, a_t^{k-1}\}, o_t^{k+1}, \dots, o_t^n\}$. Equivalently, $\boldsymbol{m}_t^{-k}$ can be written as $\{\boldsymbol{o_t}^{-k}, \boldsymbol{a}_t^{1:k-1}\}$, where $\boldsymbol{o_t}^{-k}$ denotes the joint observations of all agents except $k$, and $\boldsymbol{a}_t^{1:k-1}$ denotes the joint actions of agents 1 to $k - 1$. For the agent at the first level (*i.e.,* $k = 1$), $\boldsymbol{a}_t^{1:k-1} = \varnothing$. Then, the agent determines its action $a_t^k$ sampled from its policy $\pi_k(\cdot|o_t^k, \boldsymbol{m}_t^{-k})$ or equivalently $\pi_k(\cdot|\boldsymbol{o}_t, \boldsymbol{a}_t^{1:k-1})$ and sends it to the lower-level agents. After all agents have determined their actions, they perform the joint actions $\boldsymbol{a}_t$, which can be seen as sampled from the joint policy $\boldsymbol{\pi}(\cdot|s_t)$ *factorized* as $\prod_{k=1}^n \pi_k(\cdot|\boldsymbol{o}_t, \boldsymbol{a}_t^{1:k-1})$, in the environment and get a shared reward $r(s_t, \boldsymbol{a}_t)$ and the state transitions to next state $s'$ according to the transition probability $p(s'|s_t, \boldsymbol{a}_t)$. All agents aim to maximize the expected return $\sum_{t=0}^\infty \gamma^t r_t$, where $\gamma$ is the discount factor. The state-value function and action-value function of the level-$k$ agent are defined as follows:

$$V_{\pi_k}(s, \boldsymbol{a}^{1:k-1}) \triangleq \mathbb{E}_{\substack{s_{1:\infty} \\ \boldsymbol{a}_0^{k:n} \sim \boldsymbol{\pi}_{k:n} \\ \boldsymbol{a}_{1:\infty} \sim \boldsymbol{\pi}}} \left[ \sum_{t=0}^\infty \gamma^t r_t | s_0 = s, \boldsymbol{a}_0^{1:k-1} = \boldsymbol{a}^{1:k-1} \right]$$

$$Q_{\pi_k}(s, \boldsymbol{a}^{1:k}) \triangleq \mathbb{E}_{\substack{s_{1:\infty} \\ \boldsymbol{a}_0^{k+1:n} \sim \boldsymbol{\pi}_{k+1:n} \\ \boldsymbol{a}_{1:\infty} \sim \boldsymbol{\pi}}} \left[ \sum_{t=0}^\infty \gamma^t r_t | s_0 = s, \boldsymbol{a}_0^{1:k} = \boldsymbol{a}^{1:k} \right].$$

For the setting of multi-agent sequential decision-making discussed above, we have the following proposition.

**Proposition 1.** *If all the agents update its policy with individual TRPO (Schulman et al., 2015) sequentially in multi-agent sequential decision-making, then the joint policy of all agents is guaranteed to improve monotonically and converge.*

*Proof.* The proof is given in Appendix A. □

Proposition 1 indicates that SeqComm has the performance guarantee regardless of the priority of decision-making in multi-agent sequential decision-making. However, the priority of decision-making indeed affects the optimality of the converged joint policy, and we have the following claim.

**Claim 1.** *The different priorities of decision-making affect the optimality of the convergence of the learning algorithm due to the relative overgeneralization problem.*

We use a one-step matrix game as an example, as illustrated in Figure 1(a), to demonstrate the influence of the priority of decision-making on the learning process. Due to relative overgeneralization (Wei et al., 2018), agent $B$ tends to choose $b_2$ or $b_3$. Specifically, $b_2$ or $b_3$ in the suboptimal equilibrium is a better choice than $b_1$ in the optimal equilibrium when matched with arbitrary actions from agent $A$. Therefore, as shown in Figure 1(b), $B \rightarrow A$ (*i.e.*, agent $B$ makes decisions before $A$, and $A$'s policy conditions on the action of $B$) and *Simultaneous* (*i.e.*, two agents make decisions simultaneously and independently) are easily trapped into local optima. However, things can be different if agent $A$ goes first, as $A \rightarrow B$ achieves the optimum. As long as agent $A$ does not suffer from relative overgeneralization, it can help agent $B$ get rid of local optima by narrowing down the search space of $B$. Besides, a policy that determines the priority of decision-making can be learned under the guidance of the state-value function, denoted as *Learned*. It obtains better performance than $B \rightarrow A$ and *Simultaneous*, which indicates that dynamically determining the order during policy learning can be beneficial as we do not know the optimal priority in advance.

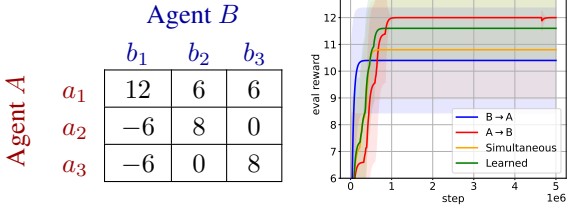

(a) payoff matrix of the game       (b) evaluations of different methods

Figure 1: (a) Payoff matrix for a one-step game. There are multiple local optima. (b) Evaluations of different methods for the game in terms of the mean reward and standard deviation of ten runs. $A \rightarrow B$, $B \rightarrow A$, *Simultaneous*, and *Learned* represent that agent $A$ makes decisions first, agent $B$ makes decisions first, two agents make decisions simultaneously, and there is another learned policy determining the priority of decision making, respectively. MAPPO (Yu et al., 2021) is used as the backbone.

**Remark 1.** The priority (*i.e.*, order) of decision-making affects the optimality of the converged joint policy in multi-agent sequential decision-making, thus it is critical to determine the order. However, learning the order directly requires an additional centralized policy in execution, which is not generalizable in the scenario where the number of agents varies. Moreover, its learning complexity exponentially increases with the number of agents, making it infeasible in many cases.

## 4    SEQUENTIAL COMMUNICATION

In this paper, we cast our eyes in another direction and resort to the world model. Ideally, we can randomly sample candidate order sequences, evaluate them under the world model (see Section 4.1), and choose the order sequence that is deemed the most promising under the true dynamic. SeqComm is designed based on this principle to determine the priority of decision-making via communication.

SeqComm adopts a multi-round communication mechanism, *i.e.*, agents are allowed to communicate with others in multiple rounds. Importantly, communication is separated into phases serving different purposes. One is the *negotiation* phase for agents to determine the priority of decision-making. Another is the *launching* phase for agents to act conditioning on actual actions upper-level agents will take to implement *explicit coordination via communication*. The overview of SeqComm is illustrated in Figure 2. Each SeqComm agent consists of a policy, a critic, and a world model, as illustrated in Figure 3, and the parameters of all networks are shared across agents (Gupta et al., 2017).

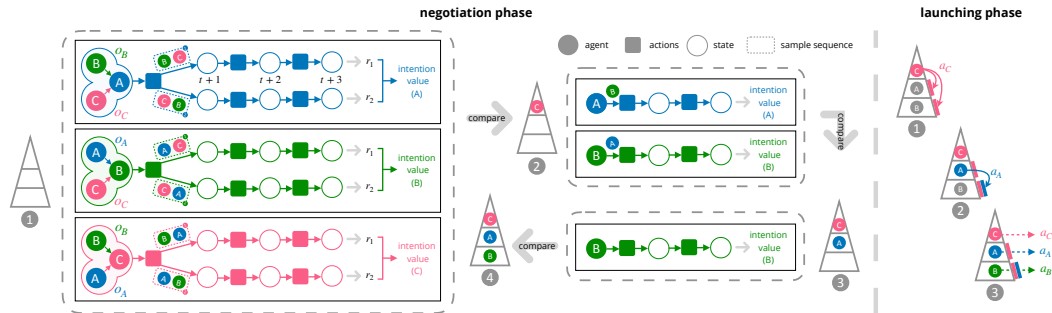

Figure 2: Overview of SeqComm. SeqComm has two communication phases, the negotiation phase (*left*) and the launching phase (*right*). In the negotiation phase, agents communicate hidden states of observations with others and obtain their own intention. The priority of decision-making is determined by sharing and comparing the value of all the intentions. In the launching phase, the agents who hold the upper-level positions will make decisions prior to the lower-level agents. Besides, their actions will be shared with anyone that has not yet made decisions.

## 4.1 NEGOTIATION PHASE

In the negotiation phase, the observation encoder first takes $o_t$ as input and outputs a hidden state $h_t$, which is used to communicate with others. Agents then determine the priority of decision-making by *intention* which is established and evaluated based on the world model.

**World Model.** The world model is needed to predict and evaluate future trajectories. SeqComm, unlike previous works (Kim et al., 2021; Du et al., 2021; Pretorius et al., 2021), can utilize received hidden states of other agents in the first round of communication to model more precise environment dynamics for the explicit coordination in the next round of communication. Once an agent can access other agents' hidden states, it shall have adequate information to estimate their actions since all agents are homogeneous and parameter-sharing. Therefore, the world model $\mathcal{M}(\cdot)$ takes as input the joint hidden states $\boldsymbol{h}_t = \{h_t^1, \ldots, h_t^n\}$ and actions $\boldsymbol{a}_t$, and predicts the next joint observations and reward,

$$\hat{\boldsymbol{o}}_{t+1}, \hat{r}_{t+1} = \mathcal{M}_i(\mathrm{AM_w}(\boldsymbol{h}_t, \boldsymbol{a}_t)),$$

where $\mathrm{AM_w}$ is the attention module. The reason that we adopt the attention module is to entitle the world model to be generalizable in the scenarios where additional agents are introduced or existing agents are removed.

**Priority of Decision-Making.** The intention is the key element to determine the priority of decision-making. The notion of intention is described as an agent's future behavior in previous works (Rabinowitz et al., 2018; Raileanu et al., 2018; Kim et al., 2021). However, we define the *intention* as an agent's future behavior *without considering others*.

As mentioned before, an agent's intention considering others can lead to circular dependencies and cause miscoordination. By our definition, the intention of an agent should be depicted as all future trajectories considering that agent as the first-mover and ignoring the others. However, there are many possible future trajectories as the priority of the rest agents is *unfixed*. In practice, we use the Monte Carlo method to evaluate intention.

Taking agent $i$ at timestep $t$ to illustrate, it firstly considers itself as the first-mover and produces its action only based on the joint hidden states, $\hat{a}_t^i \sim \pi_i(\cdot|\mathrm{AM_a}(\boldsymbol{h}_t))$, where we again use an

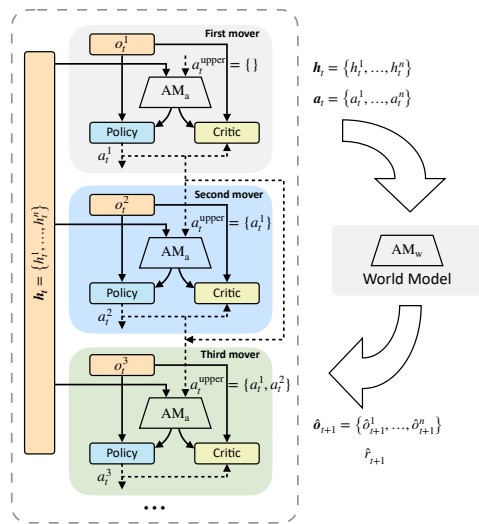

Figure 3: Architecture of SeqComm. The critic and policy of each agent take input as its own observation and received messages. The world model takes as input the joint hidden states and predicted joint actions.

attention module $\mathrm{AM_a}$ to handle the input. For the order sequence of lower-level agents, we randomly sample a set of order sequences from unfixed agents. Assume agent $j$ is the second-mover, agent $i$ models $j$'s action by considering the upper-level action following its own policy $\hat{a}_t^j \sim \pi_i(\cdot|\mathrm{AM_a}(\boldsymbol{h}_t, \hat{a}_t^i))$. The same procedure is applied to predict the actions of all other agents following the sampled order sequence. Based on the joint hidden states and predicted actions, the next joint observations $\hat{\boldsymbol{o}}_{t+1}$ and corresponding reward $\hat{r}_{t+1}$ can be predicted by the world model. The length of the predicted future trajectory is $H$ and it can then be written as $\tau^t = \{\hat{\boldsymbol{o}}_{t+1}, \hat{\boldsymbol{a}}_{t+1}, \dots, \hat{\boldsymbol{o}}_{t+H}, \hat{\boldsymbol{a}}_{t+H}\}$ by repeating the procedure aforementioned and the value of one trajectory is defined as the return of that trajectory $v_{\tau^t} = \sum_{t'=t+1}^{t+H} \gamma^{t'-t-1} \hat{r}_{t'}/H$. In addition, the intention value is defined as the average value of $F$ future trajectories with different sampled order sequences. The choice of $F$ is a tradeoff between the computation overhead and the accuracy of the estimation.

After all the agents have computed their own intention and the corresponding value, they again communicate their intention values to others. Then agents would compare and choose the agent with the highest intention value to be the first-mover. The priority of lower-level decision-making follows the same procedure with the upper-level agents fixed. Note that some agents are required to communicate intention values with others multiple times until the priority of decision-making is finally determined.

## 4.2 Launching Phase

As for the launching phase, agents communicate for obtaining additional information to make decisions. Apart from the received hidden states from the last phase, we allow agents to get what *actual* actions the upper-level agents will take in execution, while other studies can only infer others' actions by opponent modeling (Rabinowitz et al., 2018; Raileanu et al., 2018) or communicating intentions (Kim et al., 2021). Therefore, miscoordination can be naturally avoided and a better cooperation strategy is possible since lower-level agents can adjust their behaviors accordingly. A lower-level agent $i$ make a decision following the policy $\pi_i(\cdot|\mathrm{AM_a}(\boldsymbol{h}_t, \boldsymbol{a}_t^{upper}))$, where $\boldsymbol{a}_t^{upper}$ means received actual actions from all upper-level agents. As long as the agent has decided its action, it will send its action to all other lower-level agents by the communication channel. *Note that the actions are executed simultaneously and distributedly in execution, though agents make decisions sequentially.*

**Communication Overhead.** Two communication phases alternate until all agents determine their levels and get upper-level actions. Note that many previous works also adopt the multi-round communication scheme (Das et al., 2019; Singh et al., 2019). As for implementation in practice, compared with communicating high-dimensional hidden states/observations by multiple rounds (Das et al., 2019; Singh et al., 2019), or transferring multi-step trajectory (Kim et al., 2021), SeqComm needs more rounds, but it only transmits hidden states for one time. For the rest $n-1$ round communication with total $(n-1)/2$ broadcasts per agent, only a single intention value and an action will be exchanged. Considering there are $n!$ permutations of different order choices for $n$ agents, our method has greatly reduced computation overhead since each agent needs to calculate up to $n$ times to search for a satisfying order. Although SeqComm is more suitable for latency-tolerate MARL tasks, *e.g.,* power dispatch (minutes) (Wang et al., 2021a), inventory management (hours) (Feng et al., 2021), maritime transportation (days) (Li et al., 2019), it is possible for SeqComm to have a wider range of applications given the rapid development of the communication technology, *e.g.,* 5G.

## 4.3 Theoretical Analysis

As the priority of decision-making is determined by intention values, SeqComm is likely to choose different orders *at different timesteps* during training. However, we have the following proposition that theoretically guarantees the performance of the learned joint policy under SeqComm.

**Proposition 2.** *The monotonic improvement and convergence of the joint policy in SeqComm are independent of the priority of decision-making of agents at each timestep.*

*Proof.* The proof is given in Appendix A. □

The priority of decision-making is chosen under the world model, thus the compounding errors in the world model can result in discrepancies between the predicted returns of the same order under the

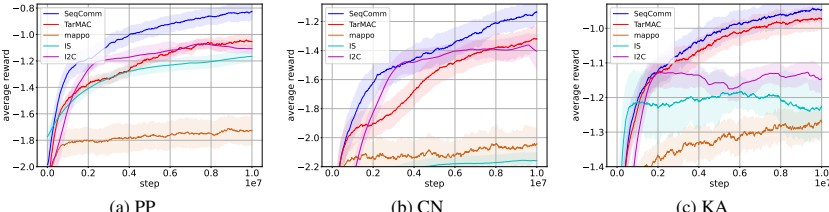

Figure 4: Learning curves in terms of the mean reward averaged over timesteps of SeqComm and baselines on three MPE tasks: (a) predator-prey, (b) cooperative navigation, and (c) keep-away.

world model and the true dynamics. We then analyze the monotonic improvement for the joint policy under the world model based on Janner et al. (2019).

**Theorem 1.** *Let the expected total variation between two transition distributions be bounded at each timestep as $\max_t \mathbb{E}_{s \sim \boldsymbol{\pi}_{\beta,t}}[D_{TV}(p(s'|s, \boldsymbol{a})||\hat{p}(s'|s, \boldsymbol{a}))] \leq \epsilon_m$, and the policy divergences at level $k$ be bounded as $\max_{s, \boldsymbol{a}^{1:k-1}} D_{TV}(\pi_{\beta,k}(a^k|s, \boldsymbol{a}^{1:k-1})||\pi_k(a^k|s, \boldsymbol{a}^{1:k-1})) \leq \epsilon_{\pi_k}$, where $\boldsymbol{\pi}_\beta$ is the data collecting policy for the model and $\hat{p}(s'|s, \boldsymbol{a})$ is the transition distribution under the model. Then the model return $\hat{\eta}$ and true return $\eta$ of the policy $\boldsymbol{\pi}$ are bounded as:*

$$\hat{\eta}[\boldsymbol{\pi}] \geq \eta[\boldsymbol{\pi}] - \underbrace{[\frac{2\gamma r_{\max}(\epsilon_m + 2\sum_{k=1}^n \epsilon_{\pi_k})}{(1-\gamma)^2} + \frac{4r_{\max}\sum_{k=1}^n \epsilon_{\pi_k}}{(1-\gamma)}]}_{C(\epsilon_m, \boldsymbol{\epsilon}_{\pi_{1:n}})}$$

*Proof.* The proof is given in Appendix B. □

**Remark 2.** Theorem 1 provides a useful relationship between the compounding errors and the policy update. As long as we improve the return under the true dynamic by more than the gap, $C(\epsilon_m, \boldsymbol{\epsilon}_{\pi_{1:n}})$, we can guarantee the policy improvement under the world model. If no such policy exists to overcome the gap, it implies the model error is too high, that is, there is a large discrepancy between the world model and true dynamics. Thus the order sequence obtained under the world model is not reliable. Such an order sequence is almost the same as a random one. Though a random order sequence also has the theoretical guarantee of Proposition 2, we will show in Section 5.2 that a random order sequence leads to a poor local optimum empirically.

## 5 EXPERIMENTS

Sequential communication (SeqComm) is currently instantiated based on MAPPO (Yu et al., 2021). We evaluate SeqComm on three tasks in multi-agent particle environment (MPE) (Lowe et al., 2017) and four maps in StarCraft multi-agent challenge (SMAC) (Samvelyan et al., 2019).

For these experiments, we compare SeqComm against the following communication-free and communication-based baselines: MAPPO (Yu et al., 2021), QMIX (Rashid et al., 2018), IS (Kim et al., 2021), TarMAC (Das et al., 2019), and I2C (Ding et al., 2020). In more detail, IS communicates predicted future trajectories (observations and actions), and predictions are made by the environment model. TarMAC uses the attention model to focus more on important incoming messages (the hidden states of observations). TarMAC is reproduced based on MAPPO instead of A2C in the original paper for better performance. I2C infers one-to-one communication to reduce the redundancy of messages (also conditioning on observations).

In the experiments, all the methods are parameter-sharing for fast convergence. We have fine-tuned the baselines for a fair comparison. Please refer to Appendix E for experimental settings and Appendix F for implementation details. All results are presented in terms of the mean and standard deviation of five runs with different random seeds.

### 5.1 RESULTS

**MPE.** We experiment on predator-prey (PP), cooperative navigation (CN), and keep-away (KA) in MPE. In PP, five predators (agents) try to capture three prey. In CN, five agents try to occupy five landmarks. In KA, three attackers (agents) try to occupy three landmarks, however, there are three

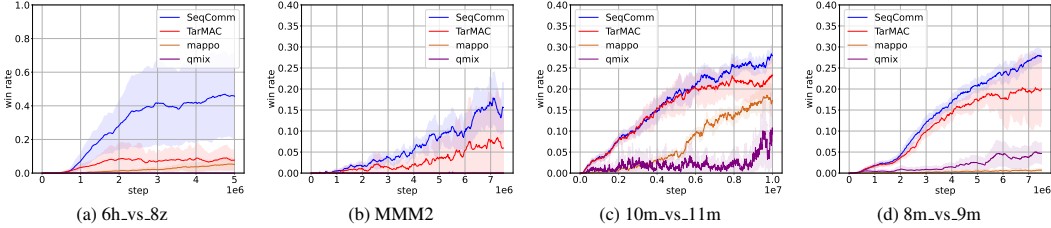

Figure 5: Learning curves in terms of the win rate of SeqComm and baselines on four customized SMAC maps: (a) 6h_vs_8z, (b) MMM2, (c) 10m_vs_11m, and (d) 8m_vs_9m.

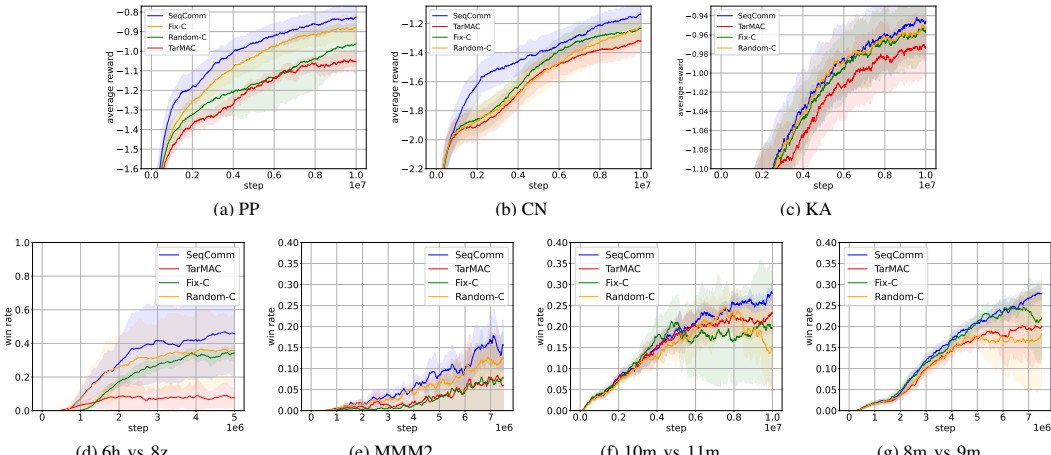

Figure 6: Ablation studies on the priority of decision-making in all the tasks. Fix-C: the priority of decision-making is fixed throughout one episode. Random-C: the priority of decision-making is determined randomly. TarMAC is also compared as a reference without explicit action coordination.

defenders to push them away. In all three tasks, the size of agents is set to be larger than the original settings so that collisions occur more easily, following the settings in (Kim et al., 2021). In addition, agents cannot observe any other agents, and this makes the task more difficult and communication more important. We can observe similar modifications in previous works (Foerster et al., 2016; Ding et al., 2020). After all, we want to demonstrate the superior over communication-based baselines, and communication-based methods are more suitable for scenarios with limited vision. More details about experimental settings are available in Appendix E.

Figure 4 shows the learning curves of all the methods in terms of the mean reward averaged over timesteps in PP, CN, and KA. We can see that SeqComm converges to the highest mean reward compared with all the baselines. The results demonstrate the superiority of SeqComm. In more detail, all communication-based methods outperform MAPPO, indicating the necessity of communication in these difficult tasks. Apart from MAPPO, IS performs the worst since it may access inaccurate predicted information due to the circular dependencies. The substantial improvement SeqComm over I2C and TarMAC is attributed to that SeqComm allows agents to get more valuable action information for explicit coordination. The agents learned by SeqComm show sophisticated coordination strategies induced by the priority of decision-making, which can be witnessed by the visualization of agent behaviors. More details are given in Appendix C. Note that QMIX is omitted in the comparison for clear presentation since Yu et al. (2021) have shown QMIX and MAPPO exhibit similar performance in various MPE tasks.

**SMAC.** We also evaluate SeqComm against the baselines on four customized maps in SMAC: 6h_vs_8z, MMM2, 10m_vs_11m, and 8m_vs_9m, where we have made some minor changes to the observation part of agents to make it more difficult. Specifically, the sight range of agents is reduced from 9 to 2, and agents cannot perceive any information about their allies even if they are within the sight range. NDQ (Wang et al., 2020) adopts a similar change to increase the difficulty of action coordination and demonstrates that the miscoordination problem is widespread in multi-agent learning. The rest settings remain the same as the default.

The learning curves of SeqComm and the baselines in terms of the win rate are illustrated in Figure 5. IS and I2C fail in this task and get a zero win rate because these two methods are built on MADDPG. However, MADDPG cannot work well in SMAC, especially when we reduce the sight range of agents, which is also supported by other studies (Papoudakis et al., 2021). SeqComm and TarMAC converge to better performances than MAPPO and QMIX, which demonstrates the benefit of communication. Moreover, SeqComm outperforms TarMAC, which again verifies the gain of explicit action coordination.

## 5.2 ABLATION STUDIES

**Priority of Decision-Making.** We compare SeqComm with two ablation baselines with only a difference in the priority of decision-making: the priority of decision-making is fixed throughout one episode, denoted as Fix-C, and the priority of decision-making is determined randomly at each timestep, denoted as Random-C. TarMAC is also compared as a reference without explicit action coordination.

As depicted in Figure 6, SeqComm achieves a higher mean reward or win rate than Fix-C, Random-C, and TarMAC in all the tasks. These results verify the importance of the priority of decision-making and the necessity to continuously adjust it during one episode. It is also demonstrated that SeqComm can provide a proper priority of decision-making. As discussed in Section 4.3, although Fix-C and Random-C also have the theoretical guarantee, they converge to poor local optima in practice. Moreover, Fix-C and Random-C show better performance than TarMAC in most tasks. This result accords with the hypothesis that the SE is likely to be Pareto superior to the average NE in games with a high cooperation level. Additionally, the learned policy of SeqComm can generalize well to the same task with a different number of agents in MPE, which is detailed in Appendix C.

**Communication Range.** We also carry out ablation studies on communication range in MPE tasks. Note that communication range means how many nearest neighbors each agent is allowed to communicate with, following the setting in Ding et al. (2020). We reduce the communication range of SeqComm from 4 to 2 and 0. As there are only three agents in KA, it is omitted in this study. The results are shown in Figure 7. Communication-based agents perform better than communication-free agents, which accords with the results of many previous studies. More importantly, the superiority of SeqComm with communication range 2 over the corresponding TarMAC again demonstrates the effectiveness of sequential communication even in reduced communication ranges.

However, as the communication range decreases from 4 to 2, there is no performance reduction in these two MPE tasks. On the contrary, the agents with communication range 2 perform the best. It accords with the results in I2C (Ding et al., 2020) and ATOC (Jiang & Lu, 2018) that redundant information can impair the learning process sometimes. In other settings, this conclusion might not be true. Moreover, since under our communication scheme agents can obtain more information, *i.e.,* the actual actions of others, it is more reasonable that SeqComm can still outperform other methods in reduced communication ranges.

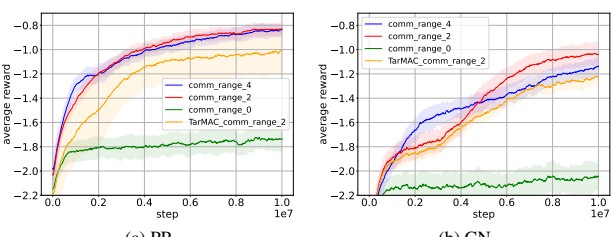

Figure 7: Ablation studies on reduced communication range in (a) predator-prey and (b) cooperative navigation.

## 6 CONCLUSIONS

We have proposed SeqComm, which enables agents explicitly coordinate with each other. SeqComm from an asynchronous perspective allows agents to make decisions sequentially. A two-phase communication scheme has been adopted for determining the priority of decision-making and communicating messages accordingly. Theoretically, we prove the policies learned by SeqComm are guaranteed to improve monotonically and converge. Empirically, it is demonstrated that SeqComm outperforms baselines in a variety of cooperative multi-agent tasks and SeqComm can provide a proper priority of decision-making.

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

# A PROOFS OF PROPOSITION 1 AND PROPOSITION 2

**Lemma 1** (Agent-by-Agent PPO). *If we update the policy of each agent $i$ with TRPO Schulman et al. (2015) (or approximately PPO) when fixing all the other agent's policies, then the joint policy will improve monotonically.*

*Proof.* We consider the joint surrogate objective in TRPO $L_{\boldsymbol{\pi}_{\text{old}}}(\boldsymbol{\pi}_{\text{new}})$ where $\boldsymbol{\pi}_{\text{old}}$ is the joint policy before updating and $\boldsymbol{\pi}_{\text{new}}$ is the joint policy after updating.

Given that $\pi_{\text{new}}^{-i} = \pi_{\text{old}}^{-i}$, we have:

$$
\begin{aligned}
L_{\boldsymbol{\pi}_{\text{old}}}(\boldsymbol{\pi}_{\text{new}}) &= \mathbb{E}_{a \sim \boldsymbol{\pi}_{\text{new}}}[A_{\boldsymbol{\pi}_{\text{old}}}(s, \boldsymbol{a})] \\
&= \mathbb{E}_{a \sim \boldsymbol{\pi}_{\text{old}}}\Big[\frac{\boldsymbol{\pi}_{\text{new}}(\boldsymbol{a}|s)}{\boldsymbol{\pi}_{\text{old}}(\boldsymbol{a}|s)} A_{\boldsymbol{\pi}_{\text{old}}}(s, \boldsymbol{a})\Big] \\
&= \mathbb{E}_{a \sim \boldsymbol{\pi}_{\text{old}}}\Big[\frac{\pi_{\text{new}}^{i}(a^i|s)}{\pi_{\text{old}}^{i}(a^i|s)} A_{\boldsymbol{\pi}_{\text{old}}}(s, \boldsymbol{a})\Big] \\
&= \mathbb{E}_{a^i \sim \pi_{\text{old}}^{i}}\left[\frac{\pi_{\text{new}}^{i}(a^i|s)}{\pi_{\text{old}}^{i}(a^i|s)} \mathbb{E}_{a^{-i} \sim \pi_{old}^{-i}}[A_{\boldsymbol{\pi}_{\text{old}}}(s, a^i, a^{-i})]\right] \\
&= \mathbb{E}_{a^i \sim \pi_{\text{old}}^{i}}\left[\frac{\pi_{\text{new}}^{i}(a^i|s)}{\pi_{\text{old}}^{i}(a^i|s)} A_{\boldsymbol{\pi}_{\text{old}}}^{i}(s, a^i)\right] \\
&= L_{\pi_{\text{old}}^{i}}(\pi_{\text{new}}^{i}),
\end{aligned}
$$

where $A_{\boldsymbol{\pi}_{\text{old}}}^{i}(s, a^i) = \mathbb{E}_{a^{-i} \sim \pi_{\text{old}}^{-i}}[A_{\boldsymbol{\pi}_{\text{old}}}(s, a^i, a^{-i})]$ is the individual advantage of agent $i$, and the third equation is from the condition $\pi_{\text{new}}^{-i} = \pi_{\text{old}}^{-i}$.

With the result of TRPO, we have the following conclusion:

$$
\begin{aligned}
J(\pi_{\text{new}}) - J(\pi_{\text{old}}) &\geq L_{\boldsymbol{\pi}_{\text{old}}}(\boldsymbol{\pi}_{\text{new}}) - C D_{\text{KL}}^{\max}(\boldsymbol{\pi}_{\text{new}} || \boldsymbol{\pi}_{\text{old}}) \\
&= L_{\pi_{\text{old}}^{i}}(\pi_{\text{new}}^{i}) - C D_{\text{KL}}^{\max}(\pi_{\text{new}}^{i} || \pi_{\text{old}}^{i}) \quad (\text{from } \pi_{\text{new}}^{-i} = \pi_{\text{old}}^{-i})
\end{aligned}
$$

This means the individual objective is the same as the joint objective so the monotonic improvement is guaranteed. $\square$

Then we can show the proof of Proposition 1.

*Proof.* We will build a new MDP $\tilde{M}$ based on the original MDP. We keep the action space $\tilde{A} = A = \times_{i=1}^{n} A^i$, where $A^i$ is the original action space of agent $i$. The new state space contains multiple layers. We define $\tilde{S}^k = S \times (\times_{i=1}^{k} A^i)$ for $k = 1, 2, \cdots, n-1$ and $\tilde{S}^0 = S$, where $S$ is the original state space. Then a new state $\tilde{s}^k \in \tilde{S}^k$ means that $\tilde{s}^k = (s, a^1, a^2, \cdots, a^k)$. The total new state space is defined as $\tilde{S} = \cup_{i=0}^{n-1} \tilde{S}^i$. Next we define the transition probability $\tilde{P}$ as following:

$$
\tilde{P}(\tilde{s}'|\tilde{s}^k, a^{k+1}, a^{-(k+1)}) = \mathbb{1}\left(\tilde{s}' = (\tilde{s}^k, a^{k+1})\right), \ k < n-1
$$

$$
\tilde{P}(\tilde{s}'|\tilde{s}^k, a^{k+1}, a^{-(k+1)}) = \mathbb{1}\left(\tilde{s}' \in \tilde{S}^0\right) P(\tilde{s}'|\tilde{s}^k, a^{k+1}), \ k = n-1.
$$

This means that the state in the layer $k$ can only transition to the state in the layer $k+1$ with the corresponding action, and the state in the layer $n-1$ will transition to the layer 0 with the probability $P$ in the original MDP. The reward function $\tilde{r}$ is defined as following:

$$
\tilde{r}(\tilde{s}, \boldsymbol{a}) = \mathbb{1}\left(\tilde{s} \in \tilde{S}_0\right) r(\tilde{s}, \boldsymbol{a}).
$$

This means the reward is only obtained when the state in layer 0 and the value is the same as the original reward function. Now we obtain the total definition of the new MDP $\tilde{M} = \{\tilde{S}, \tilde{A}, \tilde{P}, \tilde{r}, \gamma\}$.

Then we claim that if all agents learn in multi-agent sequential decision-making by PPO, they are actually taking agent-by-agent PPO in the new MDP $\tilde{M}$. To be precise, one update of multi-agent

sequential decision-making in the original MDP $M$ equals to a round of update from agent 1 to agent $n$ by agent-by-agent PPO in the new MDP $\tilde{M}$. Moreover, the total reward of a round in the new MDP $\tilde{M}$ is the same as the reward in one timestep in the original MDP $M$. With this conclusion and Lemma 1, we complete the proof.

$\square$

The proof of Proposition 2 can be seen as a corollary of the proof of Proposition 1.

*Proof.* From Lemma 1 we know that the monotonic improvement of the joint policy in the new MDP $\tilde{M}$ is guaranteed for each update of one single agent's policy. So even if the different round of updates in the new MDP $\tilde{M}$ is with different order of the decision-making, the monotonic improvement of the joint policy is still guaranteed. Finally, from the proof of Proposition 1, we know that the monotonic improvement in the new MDP $\tilde{M}$ equals to the monotonic improvement in the original MDP $M$. These complete the proof. $\square$

## B    PROOFS OF THEOREM 1

**Lemma 2** (TVD of the joint distributions). *Suppose we have two distribution $p_1(x,y) = p_1(x)p_1(x|y)$ and $p_2(x,y) = p_2(x)p_2(x|y)$. We can bound the total variation distance of the joint as:*

$$D_{TV}(p_1(x,y)||p_2(x,y)) \leq D_{TV}(p_1(x)||p_2(x)) + \max_x D_{TV}(p_1(y|x)||p_2(y|x))$$

*Proof.* See (Janner et al., 2019) (Lemma B.1). $\square$

**Lemma 3** (Markov chain TVD bound, time-varing). *Suppose the expected KL-divergence between two transition is bounded as $\max_t \mathbb{E}_{s \sim p_{1,t}(s)} D_{KL}(p_1(s'|s)||p_2(s'|s)) \leq \delta$, and the initial state distributions are the same $p_{1,t=0}(s) = p_{2,t=0}(s)$. Then the distance in the state marginal is bounded as:*

$$D_{TV}(p_{1,t}(s)||p_{2,t}(s)) \leq t\delta$$

*Proof.* See (Janner et al., 2019) (Lemma B.2). $\square$

**Lemma 4** (Branched Returns Bound). *Suppose the expected KL-divergence between two dynamics distributions is bounded as $\max_t \mathbb{E}_{s \sim p_{1,t}(s)}[D_{TV}(p_1(s'|s,\boldsymbol{a})||p_2(s'|s,\boldsymbol{a}))]$, and the policy divergences at level $k$ are bounded as $\max_{s,\boldsymbol{a}^{1:k-1}} D_{TV}(\pi_1(a^k|s,\boldsymbol{a}^{1:k-1})||\pi_2(a^k|s,\boldsymbol{a}^{1:k-1})) \leq \epsilon_{\pi_k}$. Then the returns are bounded as:*

$$|\eta_1 - \eta_2| \leq \frac{2r_{\max}\gamma(\epsilon_m + \sum_{k=1}^n \epsilon_{\pi_k})}{(1-\gamma)^2} + \frac{2r_{\max}\sum_{k=1}^n \epsilon_{\pi_k}}{1-\gamma},$$

*where $r_{\max}$ is the upper bound of the reward function.*

*Proof.* Here, $\eta_1$ denotes the returns of $\boldsymbol{\pi}_1$ under dynamics $p_1(s'|s,\boldsymbol{a})$, and $\eta_2$ denotes the returns of $\boldsymbol{\pi}_2$ under dynamics $p_2(s'|s,\boldsymbol{a})$. Then we have

$$|\eta_1 - \eta_2| = |\sum_{s,\boldsymbol{a}}(p_1(s,\boldsymbol{a}) - p_2(s,\boldsymbol{a}))r(s,\boldsymbol{a})|$$

$$= |\sum_t \sum_{s,\boldsymbol{a}} \gamma^t (p_{1,t}(s,\boldsymbol{a}) - p_{2,t}(s,\boldsymbol{a}))r(s,\boldsymbol{a})|$$

$$\leq \sum_t \sum_{s,\boldsymbol{a}} \gamma^t |p_{1,t}(s,\boldsymbol{a}) - p_{2,t}(s,\boldsymbol{a})|r(s,\boldsymbol{a})$$

$$\leq r_{\max} \sum_t \sum_{s,\boldsymbol{a}} \gamma^t |p_{1,t}(s,\boldsymbol{a}) - p_{2,t}(s,\boldsymbol{a})|.$$

By Lemma 2, we get

$$
\begin{aligned}
\max_s D_{TV}(\pi_1(\boldsymbol{a}|s)||\pi_2(\boldsymbol{a}|s)) &\le \max_{s,a_1} D_{TV}(\pi_1(\boldsymbol{a}^{-1}|s,a^1)||\pi_2(\boldsymbol{a}^{-1}|s,a^1)) \\
&\quad + \max_s D_{TV}(\pi_1(a^1|s)||\pi_2(a^1|s)) \\
&\le \cdots \\
&\le \sum_{k=1}^n \max_{s,\boldsymbol{a}^{1:k-1}} D_{TV}(\pi_1(a^k|s,\boldsymbol{a}^{1:k-1})||\pi_2(a^k|s,\boldsymbol{a}^{1:k-1})) \\
&\le \sum_{k=1}^n \epsilon_{\pi_k}.
\end{aligned}
$$

We then apply Lemma 3, using $\delta = \epsilon_m + \sum_{k=1}^n \epsilon_{\pi_k}$ (via Lemma 3 and 2) to get

$$
\begin{aligned}
D_{TV}(p_{1,t}(s)||p_{2,t}(s)) &\le t\max_t E_{s\sim p_{1,t}(s)} D_{TV}(p_{1,t}(s'|s)||p_{2,t}(s'|s)) \\
&\le t\max_t E_{s\sim p_{1,t}(s)} D_{TV}(p_{1,t}(s',\boldsymbol{a}|s)||p_{2,t}(s',\boldsymbol{a}|s)) \\
&\le t(\max_t E_{s\sim p_{1,t}(s)} D_{TV}(p_{1,t}(s'|s,\boldsymbol{a})||p_{2,t}(s'|s,\boldsymbol{a})) \\
&\quad + \max_t E_{s\sim p_{1,t}(s)} \max_s D_{TV}(\boldsymbol{\pi}_{1,t}(\boldsymbol{a}|s)||\boldsymbol{\pi}_{2,t}(\boldsymbol{a}|s))) \\
&\le t(\epsilon_m + \sum_{k=1}^n \epsilon_{\pi_k})
\end{aligned}
$$

And we also get $D_{TV}(p_{1,t}(s,\boldsymbol{a})||p_{2,t}(s,\boldsymbol{a})) \le t(\epsilon_m + \sum_{k=1}^n \epsilon_{\pi_k}) + \sum_{k=1}^n \epsilon_{\pi_k}$ by Lemma 2. Thus, by plugging this back, we get:

$$
\begin{aligned}
|\eta_1 - \eta_2| &\le r_{\max} \sum_t \sum_{s,\boldsymbol{a}} \gamma^t |p_{1,t}(s,\boldsymbol{a}) - p_{2,t}(s,\boldsymbol{a})| \\
&\le 2r_{\max} \sum_t \gamma^t (t(\epsilon_m + \sum_{k=1}^n \epsilon_{\pi_k}) + \sum_{k=1}^n \epsilon_{\pi_k}) \\
&\le 2r_{\max}(\frac{\gamma(\epsilon_m + \sum_{k=1}^n \epsilon_{\pi_k}))}{(1-\gamma)^2} + \frac{\sum_{k=1}^n \epsilon_{\pi_k}}{1-\gamma})
\end{aligned}
$$

$\square$

Then we can show the proof of Theorem 1.

*Proof.* Let $\boldsymbol{\pi}_\beta$ denote the data collecting policy. We use Lemma 4 to bound the returns, but it will require bounded model error under the new policy $\boldsymbol{\pi}$. Thus, we need to introduce $\boldsymbol{\pi}_\beta$ by adding and subtracting $\eta[\boldsymbol{\pi}_\beta]$, to get:

$$
\hat\eta[\boldsymbol{\pi}] - \eta[\boldsymbol{\pi}] = \hat\eta[\boldsymbol{\pi}] - \eta[\boldsymbol{\pi}_\beta] + \eta[\boldsymbol{\pi}_\beta] - \eta[\boldsymbol{\pi}].
$$

we can bound $L_1$ and $L_2$ both using Lemma 4 by using $\delta = \sum_{k=1}^n \epsilon_{\pi_k}$ and $\delta = \epsilon_m + \sum_{k=1}^n \epsilon_{\pi_k}$ respectively, and obtain:

$$
L_1 \ge -\frac{2\gamma r_{\max} \sum_{k=1}^n \epsilon_{\pi_k}}{(1-\gamma)^2} - \frac{2r_{\max} \sum_{k=1}^n \epsilon_{\pi_k}}{(1-\gamma)}
$$

$$
L_2 \ge -\frac{2\gamma r_{\max}(\epsilon_{\pi_m} + \sum_{k=1}^n \epsilon_{\pi_k})}{(1-\gamma)^2} - \frac{2r_{\max} \sum_{k=1}^n \epsilon_{\pi_k}}{(1-\gamma)}.
$$

Adding these two bounds together yields the conclusion.

$\square$

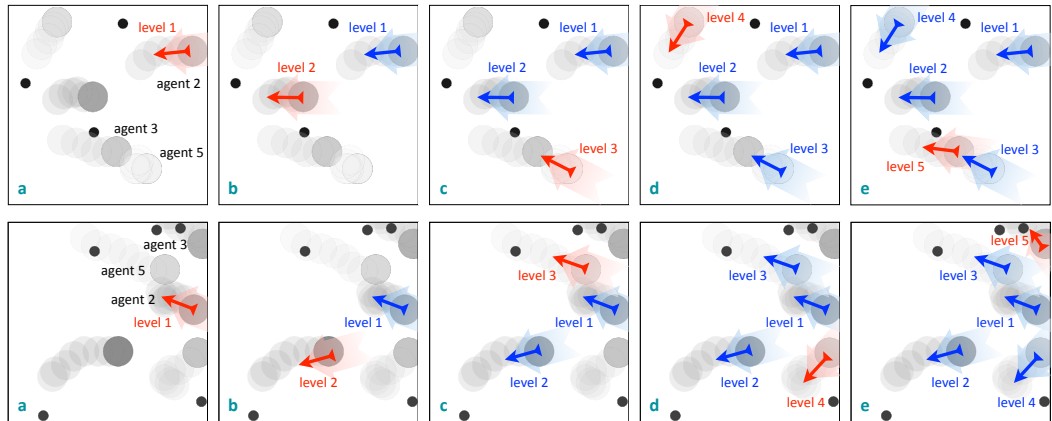

Figure 8: Illustration of learned priority of decision making in PP (*upper panel*) and CN (*lower panel*). Preys (landmarks) are viewed in black and predators (agents) are viewed in grey in PP (CN). From *a* to *e*, shown is the priority order. The smaller the level index, the higher priority of decision-making is.

Table 1: Mean reward in different tasks, averaged over timesteps, with 200 test trials.

|  | Fix-C | SeqComm |
|---|---|---|
| 3-agent in CN | $-0.83$ ±0.17 | $-\mathbf{0.76}$ ±0.08 |
| 7-agent in CN | $-1.79$ ±0.15 | $-\mathbf{1.57}$ ±0.10 |
| 7-agent in PP | $-1.89$ ±0.45 | $-\mathbf{1.31}$ ±0.60 |

# C  ADDITIONAL EXPERIMENTS

## C.1  ILLUSTRATION OF LEARNED PRIORITY OF DECISION-MAKING

Figure 8 (upper panel from $a$ to $e$) shows the priority order of decision-making determined by SeqComm in PP. Agent 2 that is far away from other preys and predators is chosen to be the first-mover. If agents want to encircle and capture the preys, the agents (*e.g.*, agent 2 and 5) that are on the periphery of the encircling circle should hold upper-level positions since they are able to decide how to narrow the encirclement. In addition, agent 3 makes decisions prior to agent 5 so that collision can be avoided after agent 5 obtains the intention of agent 3.

For CN, as illustrated in Figure 8 (lower panel from $a$ to $e$), agent 2 is far away from all the landmarks and all other agents are in a better position to occupy landmarks. Therefore, agents 2 is chosen to be the first-mover, which is similar to the phenomenon observed in PP. Once it has determined the target to occupy, other agents (agent 5 and 3) can adjust their actions accordingly and avoid conflict of goals. Otherwise, if agent 5 makes a decision first and chooses to occupy the closest landmark, then agent 2 has to approach to a further landmark which would take more steps.

## C.2  GENERALIZATION

Generalization to different numbers of agents has always been a key problem in MARL. For most algorithms in communication, once the model is trained in one scenario, it is unlikely for agents to maintain relatively competitive performance in other scenarios with different numbers of agents. However, as we employ attention modules to process communicated messages so that agents can handle messages of different lengths. In addition, the module used to determine the priority of decision-making is also not restricted by the number of agents. Thus, we investigate whether SeqComm generalizes well to different numbers of agents in CN and PP.

For both tasks, SeqComm is trained on 5-agent settings. Then, we test SeqComm in 3-agent and 7-agent settings of CN and 7-agent setting of PP. We use Fix-C trained *directly* on these test tasks to illustrate the performance of SeqComm. Note that the quantity of both landmarks and preys is

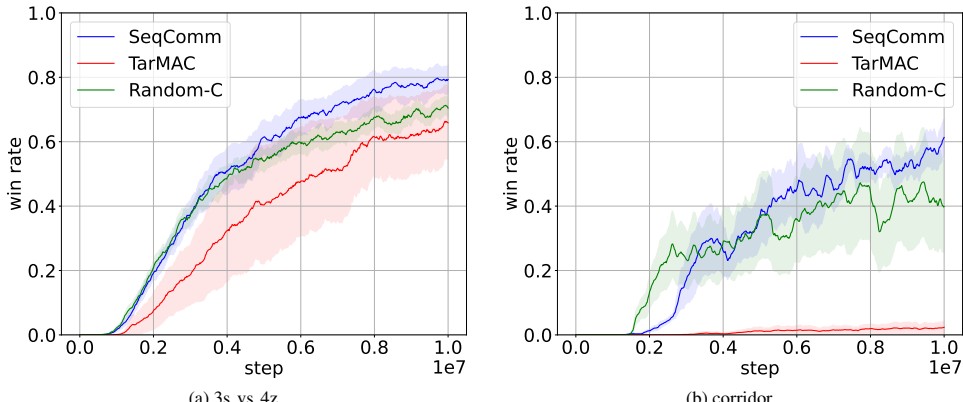

(a) 3s_vs_4z
(b) corridor

Figure 9: Learning curves in terms of the win rate of SeqComm and baselines on two customized SMAC maps: (a) 3s_vs_4z, (b) corridor.

adjusted according to the number of agents in CN and PP. The test results are shown in Table 1. SeqComm exhibits the superiority in CN and PP, demonstrating that SeqComm may have a good generalization to the number of agents. A thorough study of the generalization of SeqComm is left to future work.

### C.3 MORE SMAC MAPS

We have evaluated our method on two additional maps, *i.e.,* 3s_vs_4z and corridor. As illustrated in Figure 9, we can find out the similar conclusions as section 5.1.

## D ADDITIONAL RELATED WORK

**Multi-Agent Path Finding (MAPF).** MAPF aims to plan collision-free paths for multiple agents on a given graph from their given start vertices to target vertices. In MAPF, prioritized planning is deeply coupled with collision avoidance (Van Den Berg & Overmars, 2005; Ma et al., 2019), where collision is used to design constraints or heuristics for planning. Unlike MAPF, our method couples the priority of decision-making with the learning objective and thus is more general. In addition, the different motivations and problem settings may lead to the incompatibility of the methods in the two fields.

**Reinforcement Learning in Stackelberg Game.** Many studies (Könönen, 2004; Sodomka et al., 2013; Greenwald et al., 2003; Zhang et al., 2020) have investigated reinforcement learning in finding the Stackelberg equilibrium. Bi-AC (Zhang et al., 2020) is a bi-level actor-critic method that allows agents to have different knowledge bases so that the Stackelberg equilibrium (SE) is possible to find. The actions still can be executed simultaneously and distributedly. It empirically studies the relationship between the cooperation level and the superiority of the SE over the Nash equilibrium. AQL (Könönen, 2004) updates the Q-value by solving the SE in each iteration and can be regarded as the value-based version of Bi-AC. Existing work mainly focuses on two-agent settings and their order is fixed in advance. However, the fixed order can hardly be an optimal solution as we will show in the next section. To address this issue, we exploit agents' intentions to dynamically determine the priority of decision-making along the way of interacting with each other.

## E EXPERIMENTAL SETTINGS

In cooperative navigation, there are 5 agents and the size of each is 0.15. They need to occupy 5 landmarks with the size of 0.05. The acceleration of agents is 7. In predator-prey, the number of predators (agents) and prey is set to 5 and 3, respectively, and their sizes are 0.15 and 0.05. The acceleration is 5 for predators and 7 for prey. In keep away, the number of attackers (agents) and defenders is set to 3, and their sizes are respectively 0.15 and 0.05. Besides, the acceleration is 6

Table 2: Hyperparameters for predator-prey, cooperative navigation, keep-away

| Hyperparameter | SeqComm | Random-C | Fix-C | TarMAC | I2C | IS |
|---|---|---|---|---|---|---|
| discount ($\gamma$) | | | | 0.95,0.95,0.95 | | |
| batch size | – | – | – | – | 800 | 1024 |
| buffer capacity | – | – | – | – | $1e^6$ | |
| number of processes | | 16,16,16 | | | – | – |
| learning rate | | $1.5e^{-5}, 1e^{-5}, 4e^{-5}$ | | | $1e^{-2}, 1e^{-3}, 1e^{-3}$ | $1e^{-2}$ |
| $H$ | 10,10,20 | – | – | – | – | – |
| $F$ | | $2, 2, 1$ | – | – | – | – |

for attackers and 4 for defenders. The three landmarks are located at $(0.00, 0.30)$, $(0.25, -0.15)$, and $(-0.25, -0.15)$. Note that each agent is allowed to communicate with all other agents in all three tasks. The team reward is similar across tasks. At a timestep $t$, it can be written as $r_{\text{team}}^t = -\sum_{i=1}^n d_i^t + C^t r_{\text{collision}}$, where $d_i^t$ is the distance of landmark/prey $i$ to its nearest agent/predator, $C^t$ is the number of collisions (when the distance between two agents is less than the sum of their sizes) occurred at timestep $t$, and $r_{\text{collision}} = -1$. In addition, agents act discretely and have 5 actions (stay and move up, down, left, right). The length of each episode is 20, 30, and 20 in cooperative navigation, predator-prey, and keep-away, respectively.

# F  IMPLEMENTATION DETAILS

## F.1  ARCHITECTURE AND HYPERPARAMETERS

Our models, including SeqComm, Fix-C, and Random-C are trained based on MAPPO. The critic and policy network are realized by two fully connected layers. As for the attention module, key, query, and value have one fully connected layer each. The size of hidden layers is 100. Tanh functions are used as nonlinearity. For I2C, we use their official code with default settings of basic hyperparameters and networks. As there is no released code of IS and TarMAC, we implement IS and TarMAC by ourselves, following the instructions mentioned in the original papers (Kim et al., 2021; Das et al., 2019).

For the world model, observations and actions are firstly encoded by a fully connected layer. The output size for the observation encoder is 48, and the output size for the action encoder is 16. Then the outputs of the encoder will be passed into the attention module with the same structure aforementioned. Finally, we use a fully connected layer to decode. In these layers, Tanh is used as the nonlinearity.

Table 2 summarize the hyperparameters used by SeqComm and the baselines in the MPE.

For SMAC, SeqComm, Random-C, Fix-C are based on the same architecture, the hyperparameters stay the same. For MMM2, 6z_vs_8z, and 8m_vs_9m, the learning rate is $5e^{-5}$, while for 10m_vs_11m, corridor, and 3s_vs_4z, learning rate is $7e^{-5}$. The ppo_epoch is set to 10 for $6h\_vs\_8z$, and is 5 for rest maps. $H$ and $F$ is set to 5 and 1, respectively. However, 20 and 2 is a better value of $H$ and $F$ if computing resources is sufficient.

For TarMAC, the learning rate is $7e^{-5}$ for all maps. The ppo_epoch is set to 10 for $6h\_vs\_8z$, and is 5 for rest maps.

For MAPPO, the learning rate is $5e^{-5}$ for MMM2 and 6z_vs_8z, and $7e^{-5}$ for 8m_vs_9m and 10m_vs_11m.

For these four methods, the mini_batch is set to 1. As for other hyperparameters, we follow the default settings of the official code (Yu et al., 2021).

For QMIX, the learning rate is $5e^{-5}$. The $\epsilon$ is 1 and the batch size is 32. The buffer size is $5e^3$. For others, we follow the default settings of link https://github.com/starry-sky6688/MARL-Algorithms.git

## F.2 ATTENTION MODULE

Attention module (AM) is applied to process messages in the world model, critic network, and policy network. AM consists of three components: query, key, and values. The output of AM is the weighted sum of values, where the weight of value is determined by the dot product of the query and the corresponding key.

For AM in the world model denoted as $\mathrm{AM_w}$, agent $i$ gets messages $\boldsymbol{m}_t^{-i} = \boldsymbol{h}_t^{-i}$ from all other agents at timestep $t$ in negotiation phase, and predicts a query vector $q_t^i$ following $\mathrm{AM}_{\mathrm{w},q}^i(h_t^i)$. The query is used to compute a dot product with keys $\boldsymbol{k}_t = [k_t^1, \cdots, k_t^n]$. Note that $k_t^j$ is obtained by the message from agent $j$ following $\mathrm{AM}_{\mathrm{a},k}^i(h_t^j)$ for $j \neq i$, and $k_t^i$ is from $\mathrm{AM}_{\mathrm{neg},k}^i(h_t^i)$. Besides, it is scaled by $1/\sqrt{d_k}$ followed by a softmax to obtain attention weights $\alpha$ for each value vector:

$$\alpha_i = \mathrm{softmax}\left[ \frac{q_t^{i^T} k_t^1}{\sqrt{d_k}} \cdots \underbrace{\frac{q_t^{i^T} k_t^j}{\sqrt{d_k}}}_{\alpha_{ij}} \cdots \frac{q_t^{i^T} k_t^n}{\sqrt{d_k}} \right] \tag{1}$$

The output of attention module is defined as: $c_t^i = \sum_{j=1}^n \alpha_{ij} v_t^j$, where $v_t^j$ is obtained from messages or its own hidden state of observation following $\mathrm{AM}_{\mathrm{w},v}^i(\cdot)$.

As for AM in the policy and critic network denoted as $\mathrm{AM_a}$, agent $i$ gets additional messages from upper-level agent in the launching phase. The message from upper-level and lower-level agent can be expanded as $\boldsymbol{m}_t^{upper} = [\boldsymbol{h}_t^{upper}, \boldsymbol{a}_t^{upper}]$ and $\boldsymbol{m}_t^{lower} = [\boldsymbol{h}_t^{lower}, 0]$, respectively. In addition, the query depends on agent's own hidden state of observation $h_t^i$, but keys and values are only from messages of other agents.

## F.3 TRAINING

The training of SeqComm is an extension of MAPPO. The observation encoder $e$, the critic $V$, and the policy $\pi$ are respectively parameterized by $\theta_e$, $\theta_v$, $\theta_\pi$. Besides, the attention module $\mathrm{AM_a}$ is parameterized by $\theta_a$ and takes as input the agent's hidden state, the messages (hidden states of other agents) in the negotiation phase, and the messages (the actions of upper-level agents) in launching phase. Let $\mathcal{D} = \{\tau_k\}_{k=1}^K$ be a set of trajectories by running policy in the environment. Note that we drop time $t$ in the following notations for simplicity.

The value function is fitted by regression on mean-squared error:

$$\mathcal{L}(\theta_v, \theta_a, \theta_e) = \frac{1}{KT} \sum_{\tau \in \mathcal{D}} \sum_{t=0}^{T-1} \left\| V(\mathrm{AM_a}(e(\boldsymbol{o}), \boldsymbol{a}^{upper})) - \hat{R} \right\|_2^2 \tag{2}$$

where $\hat{R}$ is the discount rewards-to-go.

We update the policy by maximizing the PPO-Clip objective:

$$\mathcal{L}(\theta_\pi, \theta_a, \theta_e) = \frac{1}{KT} \sum_{\tau \in \mathcal{D}} \sum_{t=0}^{T-1} \min(\frac{\pi(a|\mathrm{AM_a}(e(\boldsymbol{o}), \boldsymbol{a}^{upper}))}{\pi_{old}(a|\mathrm{AM_a}(e(\boldsymbol{o}), \boldsymbol{a}^{upper}))} A_{\pi_{old}}, g(\epsilon, A_{\pi_{old}})) \tag{3}$$

where $g(\epsilon, A) = \begin{cases} (1+\epsilon)A & A \geq 0 \\ (1-\epsilon)A & A \leq 0 \end{cases}$, and $A_{\pi_{old}}(\boldsymbol{o}, \boldsymbol{a}^{upper}, a)$ is computed using the GAE method.

The world model $\mathcal{M}$ is parameterized by $\theta_w$ is trained as a regression model using the training data set $\mathcal{S}$. It is updated with the loss:

$$\mathcal{L}(\theta_w) = \frac{1}{|\mathcal{S}|} \sum_{\boldsymbol{o}, \boldsymbol{a}, \boldsymbol{o}', r \in \mathcal{S}} \left\| (\boldsymbol{o}', r) - \mathcal{M}(\mathrm{AM_w}(e(\boldsymbol{o}), \boldsymbol{a})) \right\|_2^2. \tag{4}$$

We trained our model on one GeForce GTX 1050 Ti and Intel(R) Core(TM) i9-9900K CPU @ 3.60GHz.

