# OpenReview forum: "Multi-Agent Sequential Decision-Making via Communication"
_ICLR.cc/2023/Conference — Submitted to ICLR 2023_

### Official Review · Reviewer_7QwV · 2022-10-23

**Confidence:** 2
**Correctness:** 4
**Technical Novelty And Significance:** 2
**Empirical Novelty And Significance:** 2
**Recommendation:** 6

**Clarity, Quality, Novelty And Reproducibility:**

The experiments are comprehensive and the results look sound and presented in sufficient detail and clarity. Experiment settings are described clearly.

**Strength And Weaknesses:**

Strengths:

The problem is well-motivated. The paper is overall clearly writen and well organized. The idea of introducing communication into the process is natural and the proposed approach of hierarchical communication is also novel and looks interesting.

Weaknesses:

Though the idea of hierarchical communication is interesting, it is unclear what are its advantages compared with letting one agent makes a centralized decision and broadcasting the decision (especially now that the agents' objectives are the same and their policy space is already broken down through factorization).

**Summary Of The Paper:**

The paper studies a multiagent POMDP and proposes a communication mechanism for the agents to exchange information about their decision-making. In the process, the agents have the same objective, aiming to find a joint policy that maximizes their utility. The authors introduced a communication mechanism for the agents, which consists of a negotiation phase and a lauching pahse. In the negotiation phase, the agents communicate their observations, and in the lauching pahse, they communicate their decision-making. The latter is further implemented according to a hierarchical structure, where the agent at each level k send information to the agent at level k+1. The paper prents both theoretical results about the convergence of the agents' strategies to an equilibrium and empirical results about the performance of the proposed approach.

**Summary Of The Review:**

In summary, a well presented paper with an interesting key idea and comprehensive results. Though the advantages of the proposed approach are a bit unclear compared with centralized decision-making (see Weaknesses) and require more justifications.

---

> ### Author Response · Authors · 2022-11-11
> **Responses to Reviewer 7QwV**
>
> Thank you for your insightful comments and acknowledgment of our contribution. We do hope that the reviewer can support us in the discussion period if you think this paper has provided insight that should be shared with the MARL community
>
> > it is unclear what are its advantages compared with letting one agent makes a centralized decision and broadcasting the decision (especially now that the agents' objectives are the same and their policy space is already broken down through factorization).
>
> Things can be totally different between joint learning and communication. Two situations arise for joint learning. The first is that joint learning gets the joint observations and outputs the joint actions. In this case, the search space of joint actions is extremely large and the training procedure can suffer from the curse of dimensionality. Supposing that there are 10 agents with 5 action choices, the search space is $5^10$. It is unlikely for the joint learning to converge. The seminal work VDN [a] has proved this. The centralized decision method (C Comb) performs worse in all the baselines. To our best knowledge, we can hardly find any previous communication-based works following this setting.
> [a] Value-Decomposition Networks For Cooperative Multi-Agent Learning, AAMAS 2018.

---

> > ### Comment · Reviewer_7QwV · 2022-11-23
> > **Thank you for your response**
> >
> > Thank you for your response.
> >
> > Just one comment: I think simply listing the size of the searching space isn't a convincing arguement for the claim that the problem isn't solvable. There are many combinatorial problems where the size of the search space easily go beyond $5^{10}$, yet they can be solved efficiently. Linear programming for example: if you count the number of extreme points of the feasible space, there are exponentially many of them; yet linear programming (more specifically, to find an optimal extreme point) is a tractable problem.

---

### Official Review · Reviewer_TanN · 2022-10-25

**Confidence:** 2
**Correctness:** 3
**Technical Novelty And Significance:** 2
**Empirical Novelty And Significance:** 2
**Recommendation:** 6

**Clarity, Quality, Novelty And Reproducibility:**

The paper's quality and clarity are satisfactory.
The theoretical proposal has a certain amount of novelty.
Information for implementation is provided to reproduce the experiment.

**Strength And Weaknesses:**

Strength
+ The theoretical proof is provided.
+ The performance is empirically shown in the experiment.

I think the title does not sufficiently represent this paper's main characteristics. "Asynchronicity" seems to be the keyword of the paper.

**Summary Of The Paper:**

The paper proposes a novel communication scheme, Sequential Communication (SeqComm), for cooperative multi-agent reinforcement learning (MARL).
In communication for cooperation in MARL, circular dependencies can sometimes occur. This is caused by synchronization in communication. The proposed model assumes asynchronous communication. The upper-level agents make decisions before the lower-level ones.
They, theoretically, show that policies learned by SeqComm are
guaranteed to improve performance monotonically and converge. Also, the empirical performance is shown by comparing the proposal with other existing methods.

**Summary Of The Review:**

The paper proposes a new communication method, SeqComm, for multi-agent reinforcement learning.
They provide not only empirical results but also theoretical proof as well.
The paper is properly written.

---

> ### Author Response · Authors · 2022-11-11
> **Responses to Reviewer TanN**
>
> Thanks for your insightful comments and acknowledgment of our contribution. We do hope that the reviewer can support us in the discussion period if you think this is an important topic that will be of interest to ICLR attendees.
>
> > I think the title does not sufficiently represent this paper's main characteristics. "Asynchronicity" seems to be the keyword of the paper.
>
> We will carefully consider the usage of asynchronicity in the revision version.

---

### Official Review · Reviewer_44Uf · 2022-10-26

**Confidence:** 4
**Correctness:** 2
**Technical Novelty And Significance:** 3
**Empirical Novelty And Significance:** 2
**Recommendation:** 3

**Clarity, Quality, Novelty And Reproducibility:**

Clarity
It is unclear how SeqComm would be applicable to arbitrary MARL tasks. Even if applicable, the performance gain must be normalized against the additional computation.

Quality
Plots against training steps are misleading. "Miscoordination" on page 6 is not defined, so its avoidance cannot be measured. The latency tolerance disclaimer on page 6 is unsupported. Improvements on SeqComm runtime on real-world communication technology (e.g., 5G) cannot be measured/claimed unless accompanied by training (wall clock) times in seconds.

Originality
To the best of my knowledge, SeqComm is a novel algorithm.

**Strength And Weaknesses:**

Introducing sequential decision making may prove both a valuable and feasible approach to some MARL tasks.

However, the domain on which the proposed algorithm would successfully operate seems largely exaggerated. The underlying game-theoretic construct of MARL tasks should most definitely considered as a given, and not as a control knob. Whether a game will play out sequentially or simultaneously is -- at least within the approach and scope of building a novel MARL algorithm -- the innate characteristic of the game and not a choice for the algorithm designer. SeqComm may function correctly in MARL tasks that naturally and inherently involve sequentiality, but such a demonstration is missing. This lack significantly undermines the applicability of the algorithm.

Even if there is some justification as to the applicability of SeqComm on arbitrary MARL tasks, the performance gain comes at a huge cost of communicating n-1 rounds. Actual performance may deteriorate from this best case (n-1 rounds), as "some agents are required to communicate intention values with others multiple times until the priority of decision-making is finally determined." Furthermore, each of those rounds involves Monte Carlo sampling. This means that every single tick in the x-axis (steps) actually involves, at best n-1 "sub-steps", each of which carries out Monte Carlo sampling. Since every figure in the paper is plotted against the number of training steps, not a single figure pictures a fair comparison between SeqComm and the baselines.

Moreover, the paper needs a much better foundation in terms of its connection to game theory. Simultaneous games are prone to non-stationarity precisely because the players' information sets are limited. Making a distinction between the decision-making phase and the action execution phase does not change the fact SeqComm alters information sets. There is no discussion whatsoever in this regard (e.g., information sets, bounded rationality, non-stationarity), and SeqComm proceeds straight to interfere with the players' information sets, which are dictated by the nature of the game and not by the algorithm's working mechanism.

**Summary Of The Paper:**

Authors present a sequential communication framework to address the relative overgeneralization problem in multi-agent reinforcement learning and test it against a number of communication-free and communication-based baselines. Performance figures drawn against the number of training steps show higher average rewards in the MPE and SMAC environments.

**Summary Of The Review:**

Major concerns raised above render SeqComm borderline faulty. I find it unlikely that additional experiments, clarification, and/or proofs would make the paper suitable for this venue.

---

> ### Author Response · Authors · 2022-11-11
> **Responses to Reviewer 44Uf**
>
> To begin with, we thank the reviewer for supporting of the novelty. However, we believe the reviewer has a misunderstanding with our paper. We do hope the reviewer can go through the paper again from the perspective of cooperative MARL, not from the perspective of game theory. In cooperative MARL, all agents aim to maximize team rewards, so they can communicate information, coordinate actions, etc., as long as they do not break the simultaneous execution of actions in the environment. So, information set and bounded rationality as the reviewer mentioned are not a problem at all in cooperative MARL.
>
> > However, the domain on which the proposed algorithm would successfully operate seems largely exaggerated. The underlying game-theoretic construct of MARL tasks should most definitely be considered as a given, and not as a control knob.
>
> It is important to know that we did not change the underlying game-theoretic construct of MARL tasks. They are always fully cooperative games. We mentioned in section 4.2 that actions are executed simultaneously and distributedly in execution, though agents make decisions sequentially.
>
> Normally, in the simultaneous game, agents cannot access the actions of others, however, if it is possible to know others’ actions, why not let that happen? Better decisions can be made with more information. With the help of the multi-round communication mechanism, the sequential decision is feasible in any cooperative game. That is, lower-level agents can access the actions of upper-level agents. Note that we only make a mild assumption that the actions made in the decision phase will be executed without change. We implement our method on simultaneous SMAC without changing the innate characteristic of the game and we believe there is no reason our work cannot be applied in other cooperative MARL tasks.
>
> The key point is that only under a sequential decision framework, lower-level agents can access more information, i.e., the actions of upper-level agents, so that they are able to better coordinate with them and circular dependencies will not happen.
>
> > Even if there is some justification as to the applicability of SeqComm on arbitrary MARL tasks, the performance gain comes at a huge cost of communicating n-1 rounds.
>
> Firstly, we follow the cost-free communication settings as many previous works (more details in Section 3). These works focus on issues other than communication costs. Although the real world rarely exhibits such ideal conditions, cost-free communication allows us to provide the theoretical analysis of the proposed communication framework more conveniently. Therefore, we can provide some insights of the benefit of communication.
>
> Secondly, we indeed need to communicate n-1 rounds, but only the single intention/action value will be exchanged for multiple round. The high-dimension observation only needs one transmission. It would definitely reduce the communication overhead. In many real-world applications, the MARL model is trained firstly in the simulator and then deployed in the real application. For example, we cannot train the model in the real-world environment for inventory management (hours), and maritime transportation (days). This is because there may be a long time lag between the two decision step in the real world. In addition, we mentioned 5G only for the claim that multiple-round communication is feasible in the real world.
>
> Thirdly, most work has limitations. We sacrifice the communication cost so that more information is available to agents. From our empirical results, we indeed show performance improvement with multi-round communication. It is also common in other machine learning domains that research may sacrifice the model complexity for marginal performance improvement.
>
> We are the first to propose sequential multi-round communication, we hope more tolerance should be given and we want to share some insight of the sequential decision-making with the community.
>
> As for involving Monte Carlo sampling, the monte Carlo sampling doesn’t need to interact with the real environment. The sampled trajectory is obtained from our trained world model. It should be considered an inference procedure based on the trained model. Moreover, the x-axis is the environment step, and none of the previous model-based RL methods take into account the step in the model..
>
> > Moreover, the paper needs a much better foundation in terms of its connection to game theory.
>
> Our paper is a MARL paper, not a game theory paper. As mentioned at the beginning, since all agents aim to maximize the team rewards, information sets, and bounded rationality are not a problem. As we consider simultaneous games, we do care about non-stationarity. However, Proposition 1 and 2 theoretically show that the joint policy of all agents improves monotonically and converges in sequential decision-making. This implies that non-stationarity is also addressed.

---

### Official Review · Reviewer_emu3 · 2022-10-29

**Confidence:** 4
**Correctness:** 3
**Technical Novelty And Significance:** 3
**Empirical Novelty And Significance:** 2
**Recommendation:** 5

**Clarity, Quality, Novelty And Reproducibility:**

## Clarity

The paper is fairly clearly written. I do not believe that the authors open-source their code, which makes reproducibility difficult. It is also not clear to me which implementation of the QMIX and MAPPO baselines is used or how hyperparameters were chosen for the baselines.

**Strength And Weaknesses:**

## Strengths
- The setting of adding some communication to CTDE is an interesting way to alleviate miscoordination and work in this area is welcome.
- The paper is quite clearly written.
- The authors include ablation studies to evaluate why their method works.
- The matrix game in Figure 1 nicely illustrates why an order is theoretically useful.

## Weaknesses

- I'm not sure that I understand the setting. If agents can broadcast functions of their observations to all other agents, then how is that different from joint learning, where all agents can view the same joint observation? This seems to not be the CTDE setting at all to me, but instead joint learning. I understand that TarMAC [1] adopts a similar setting, but I would appreciate some clarification from the authors on how this differs from joint learning.
- The communication-free baselines MAPPO and IPPO are not a fair comparison, and it is not clear that SeqComm's performance is better TarMAC or the random-priority ablation. The authors claim that their method clearly outperforms the ablations and TarMAC, but the gap is only slight and seems to be mostly within a standard deviation.
- The authors do not compare with a centralised method such as PPO (but conditioning the policy on the joint observation and outputting the joint action). This seems strange given this is an obvious alternative to the method, and would require approximately the same communication.
- The empirical evaluation is only over 4 SMAC maps, and no further results are included. This does not seem enough to provide convincing evidence of outperformance.

**Summary Of The Paper:**

## Summary
The paper presents SeqComm, a multi-agent communication scheme allowing agents to condition on one another's actions by imposing ordering over the agents. The paper introduces multi-agent sequential decision and demonstrates that ordering in this paradigm can affect the optimality of the learnt policy.

The authors then present SeqComm. Each agent in SeqComm learns a policy which conditions on the joint hidden state and other agents' actions via an attention mechanism. The ordering is chosen by weighing the value of each agent's intention, which is the paper defines as the agent's future behaviour without considering the action of others. In the second phase of communication, the agents the produce a joint action. The authors then prove that monotonic improvement of the policy is independent of priority, and provide a bound on the performance loss associated with using a world model for the ordering.

The authors then evaluate SeqComm on MPE and SMAC tasks.

**Summary Of The Review:**

Although some communication is no doubt useful in MARL, requiring that functions of the observation can be broadcast to all allies does not seem to be decentralised execution to me. Additionally, I remain unconvinced by the empirical results both in their breadth and performance difference.

---

> ### Author Response · Authors · 2022-11-11
> **Responses to Reviewer emu3**
>
> > I'm not sure that I understand the setting. If agents can broadcast functions of their observations to all other agents, then how is that different from joint learning, where all agents can view the same joint observation? This seems to not be the CTDE setting at all to me, but instead joint learning. I understand that TarMAC adopts a similar setting, but I would appreciate some clarification from the authors on how this differs from joint learning.
>
> Things can be totally different between joint learning and communication. Two situations arise for joint learning. The first is that joint learning gets the joint observations and outputs the joint actions. In this case, the search space of joint actions is extremely large and the training procedure can suffer from the curse of dimensionality. Supposing that there are 10 agents with 5 action choices, the search space is $5^10$. It is unlikely for the joint learning to converge. The seminal work VDN [a] has proved this. This is showed that the joint learning method (C Comb) performs worse than all other methods. To the best of our knowledge, we can hardly find any previous communication-based works following this setting.
>
> The second is that one brain agent has multiple policy heads outputting the action for each agent separately. If so, there is no difference between this framework and CTDE, since each policy head cannot use the actions of others. In other words, the problems that happen in the CTDE framework can still happen in one brain agent.
>
> In addition, we also carry out ablation studies on communication range in MPE. In this setting, agents are only allowed to communicate with nearby agents. We have shown that our method can still perform decently with constrained communication.
>
> > The communication-free baselines MAPPO and IPPO are not a fair comparison, and it is not clear that SeqComm's performance is better TarMAC or the random-priority ablation. The authors claim that their method clearly outperforms the ablations and TarMAC, but the gap is only slight and seems to be mostly within a standard deviation.
>
> We take MAAPO as the backbone, therefore the comparison is to demonstrate the necessity of communication.
>
> To be precise, our method outperforms TarMAC by more than a standard deviation in PP, CN, and 6h_vs_8h. In MMM2, 10m_vs_11m, and 8m_vs_9m, the variance of TarMAC with different random seeds is very large, showing its learning instability. Moreover, we did not claim our method clearly outperforms the random-priority ablation. We only stated in Section 5.2 second paragraph that “SeqComm achieves a higher mean reward or win rate than Random-C.” This is the fact and the reason is also given in the same paragraph.
>
> Moreover, reviewer 44UF mentioned the Performance figures drawn against the number of training steps show higher average rewards in the MPE and SMAC environments. Reviewer 7QwV mentioned the experiments are comprehensive and the results look sound and presented in sufficient detail and clarity.
>
> > The authors do not compare with a centralised method such as PPO.
>
> It is unlikely for the joint learning to converge due to the curse of dimensionality. The seminal work VDN [a] has proved this. The centralized decision method (C Comb) performs worse in all the baselines. To our best knowledge, we can hardly find any previous communication-based works following this setting.
>
> > The empirical evaluation is only over 4 SMAC maps.
>
> Note that we choose 4 hard/super hard maps with more agents. We believe these four maps are representative and challenging enough. In addition, for other easy maps or maps with fewer agents, MAPPO has already demonstrated very promising performance even with limited sight range. Therefore, the potential improvement is largely reduced.
>
> However, we are running additional experiments in SMAC and will keep updating the results.
>
> > I do not believe that the authors open-source their code, which makes reproducibility difficult. It is also not clear to me which implementation of the QMIX and MAPPO baselines is used or how hyperparameters were chosen for the baselines.
>
> We promise in the Open-Review platform that we will open-source the code once this paper is accepted. QMIX and MAPPO baselines are based on the official code. We promise that we have fine-tuned the baselines for a fair comparison.
>
> Additionally, we want to emphasize that our method aims to provide a new perspective on sequential communication. We have theoretically analyzed how the order of decision-making influences model performance and convergence. We believe the theoretical result is the key contribution of the paper. We are the first to give a theoretical analysis of sequential decision-making in the setting of more than two agents. With our theoretical analysis, the following studies are encouraged to make the model more applicable.
>
> [a] Value-Decomposition Networks For Cooperative Multi-Agent Learning, AAMAS 2018.

---

> > ### Comment · Reviewer_emu3 · 2022-11-15
> > **Response to Authors**
> >
> > Thanks very much for your response and the discussion.
> >
> > > Things can be totally different between joint learning and communication.
> >
> > I take your point that the decentralised action space is different to joint learning. However, the shared observation space means that partial
> > observability is no longer an issue -- if one agent knows a piece of information, in principle they all will. This means that the only issue that must be overcome is coordination in the action space. As far as I am aware, such coordination issues only occur if coordination over actions is required at a single timestep. This alleviates partial observability, which is a major challenge in MARL. Difficulty in coordination and the need for communication arises because of partial observability. It seems very odd to me to ignore this and still claim to be in the CTDE setting. Decentralised execution means decentralised observations *and* actions. The Dec-POMDP-Com to me seems to be equivalent to a Multi-agent MDP, where agents share observations. This is therefore essentially a single-agent problem with a factored action space. I'm still very confused about your claims to be operating in the CTDE setting and this remains my major objection in the paper.
> >
> > Additionally, I'm not sure that there is evidence that many of the benchmarks that you have chosen actually demonstrate the coordination problems you claim to be solving. Certainly SMAC has no simultaneous coordination issues -- work has explored adding a penalty to the reward for ally health [1], but that is not used here as far as I can tell. Similarly the MPE tasks do not seem to demonstrate these problems, although they have some penalty for
> > collisions . That is why I have asked for the PPO without joint actions baseline -- I am not convinced that there is actually a significant problem to solve here in these empirical tasks.
> >
> > >It is unlikely for the joint learning to converge due to the curse of dimensionality. The seminal work VDN [a] has proved this. The centralized decision method (C Comb) performs worse in all the baselines. To our best knowledge, we can hardly find any previous communication-based works following this setting.
> >
> > I should clarify what I mean here. I do not mean a naive centralised PPO, but one where the observation space is the joint observation and the agents still act individually -- ie PPO with factored actions. This algorithm would also operate in your setting and seems a logical baseline to include. The only baselines included on SMAC are TarMAC and communication-free ones.
> >
> >
> >
> > > Note that we choose 4 hard/super hard maps with more agents. We believe these four maps are representative and challenging enough. In addition, for other easy maps or maps with fewer agents, MAPPO has already demonstrated very promising performance even with limited sight range. Therefore, the potential improvement is largely reduced.
> >
> > I note that Wang et al. [2] evaluate on a different set of SMAC maps. Is there a reason that you evaluate on these 4 maps rather than those ones?
> >
> >
> > > We promise in the Open-Review platform that we will open-source the code once this paper is accepted. QMIX and MAPPO baselines are based on the official code. We promise that we have fine-tuned the baselines for a fair comparison.
> > I'm sorry but I can't just take promises -- at least list the hyperparameters considered in the appendix. Additionally one of your SMAC experiments seems to have been run for 10 million steps rather than 1 million. Is there a reason for this?
> >
> > Overall, my concerns about the setting and quality of the baselines and experimental rigour remain.
> >
> > [1] UneVEn: Universal Value Exploration for Multi-Agent Reinforcement Learning https://arxiv.org/pdf/2010.02974.pdf
> > [2] Learning Nearly Decomposable Value Functions Via Communication Minimization https://arxiv.org/abs/1910.05366

---

> > > ### Author Response · Authors · 2022-11-16
> > > **Responses to Reviewer emu3**
> > >
> > > > CTDE
> > >
> > > Most previous communication works claim this line of research follows CTDE, such as seminal work TarMAC [1] (shared observations), I2C, and IS (shared observations). Since SeqComm is a communication-based work, we are supposed to follow the definition of previous works.
> > >
> > > As the reviewer points out, decentralized execution means decentralized observations and actions.
> > > The reviewer needs references to back up the claim. Even if so, how to define execution with decentralized actions? Definitely, we cannot say it is centralized execution. Maybe, partial decentralized execution is suitable as the claim of the reviewer. Unfortunately, we cannot make up the terminology if previous works have claimed similar settings.
> > >
> > > Dec-POMDP-Com is also defined by previous works [3]. Clearly, the Dec-POMDP-Com is not equivalent to a multi-agent MDP. First, the joint observation is not equal to the state. In addition, in communication-based works, the observations of others have been compressed to messages, which may only include parts of key information of observations. Second, multi-agent MDP cannot communicate actions with others. Note that it is possible to transform the former into the latter.
> > >
> > > In SMAC, overkilling is the key coordination problem [4]. If two agents try to kill the same enemies, it is easy to cause overkilling when the health of the enemy is little. Therefore, one of the agents can turn to other enemies when it knows the intention of the other agent.
> > >
> > > In MPE, the target conflict is the key coordination problem. To illustrate, if two agents try to occupy the same landmark, miscoordination happens.
> > >
> > > > Naive centralized PPO
> > >
> > > TarMAC is such a baseline. It takes MAPPO as backbone and takes joint observations as input for each agent. The only difference is that it uses the attention module to better process observations. In addition, TarMAC shows the attention method outperforms the no-attention baselines.
> > >
> > > > SMAC maps
> > >
> > > Most maps in [2] are customized. It is not from the standard maps provided by the paper [4], the StarCraft Multi-Agent Challenge. The question should be why they choose customized maps, not the standard SMAC maps.
> > >
> > > > Open-source the code
> > >
> > > Unfortunately, we felt offended by this part. After we make a promise in the Open-Review platform where everyone can view. The reviewer still said he/she cannot take promises. It is odd to us why we would deceive the reviewer in the open-review platform.
> > >
> > > Still, we are confused why the reviewer is tangled in communication-free baseline. Note that we set the vision range small in the SMAC maps, which makes communication essential and more coordination is needed. If communication-based methods cannot outperform the baselines, this line of research is meaningless.
> > >
> > > > 10 million steps rather than 1 million.
> > >
> > > Since we make our map harder, more training steps are needed. It is fair to all the methods.
> > >
> > >
> > > [1] A Das et al., TarMAC: Targeted Multi-Agent Communication.
> > >
> > > [2] Learning Nearly Decomposable Value Functions Via Communication Minimization.
> > >
> > > [3] Claudia V. Goldman and Shlomo Zilberstein. Optimizing information exchange in cooperative
> > > multi-agent systems.
> > >
> > > [4] M Samvelyan et al., the StarCraft Multi-Agent Challenge.

---

> > > > ### Comment · Reviewer_emu3 · 2022-11-16
> > > > **Thanks very much for the response**
> > > >
> > > > > Most previous communication works claim this line of research follows CTDE, such as seminal work TarMAC [1] (shared observations), I2C, and IS (shared observations). Since SeqComm is a communication-based work, we are supposed to follow the definition of previous works.
> > > >
> > > > I appreciate that previous works make this claim, but I do not think those works meet the definition of CTDE either. The foundational works in this setting CTDE, namely COMA [1] and MADDPG [2], both consider a setting without access to the joint observation. The planning literature which predates these works, where planning is done in a centralised fashion for decentralised policies [3] [4] also use the Dec-POMDP framework. Additionally, when describing Dec-POMDPs, Olihoek and Amato [5] mention the case of broadcast communication, and state that it falls under the framework of a multi-agent POMDP ([5], page 12).
> > > >
> > > > > Dec-POMDP-Com is also defined by previous works [3].
> > > >
> > > > My apologies, I was pretty imprecise here. The framework you have described here (not the Dec-POMDP-Com, which includes costs for
> > > > communicating etc. and seems a more complicated beast) is very very similar to an MPOMDP. In fact, I think the only difference is that the historical actions for some agents are not broadcast. MPOMDPs are a special type of POMDP and so, because of the one-brain aspect, are in some sense not really multi-agent at all [5]. I'm therefore confused about the purpose of exploring this setting. You pointed out that miscoordination problems could still occur in action space, but to my eyes do not provide convincing evidence that this is a problem in the empirical settings you describe.
> > > >
> > > > >the observations of others have been compressed to messages, which may only include parts of key information of observations
> > > >
> > > > This is contradicted by the framework that you describe on page 3, where you clearly state that the messages can contain the observations and actions of other agents. Regardless, I do not believe this kind of compression is particularly difficult -- the observation spaces in the tasks considered are not that large and would not represent large communication overheads if such communication were possible.
> > > >
> > > > >In SMAC, overkilling is the key coordination problem [4].
> > > > > In MPE, the target conflict is the key coordination problem.
> > > > > TarMAC is such a baseline.
> > > >
> > > > Sure, without access to the joint observation these tasks are very difficult, but without a clear description of how you tuned the hyperparameters of TarMAC after reimplementing it, and given the general difficulty of reproducing RL results and the not enormous performance difference between SeqComm and TarMAC, it's not clear to me that the empirical claims are very strong at all.
> > > >
> > > > Secondly, the major difference between this setting and a single-agent one is that the action space is factored. This can lead to poor coordination when one agent explores and another chooses an optimal action. However, SMAC in particular does not seem to be a clear representative of these problems, and so it is not clear to me how SeqComm is being tested differently from a single-agent algorithm with a factored action space in this benchmark.
> > > >
> > > > > Since we make our map harder, more training steps are needed. It is fair to all the methods.
> > > >
> > > > I was referring to figure 5. You seem to have trained all the scenarios for 1 million steps except 10m_vs_11m, which has been trained for 10 million steps, especially given this is the easiest  scenario.  This seems like a very strange and inconsistent evaluation protocol, even if it is the same for all algorithms. This requires some serious and specific justification if it is not just a simple mistake.
> > > >
> > > > >Unfortunately, we felt offended by this part. After we make a promise in the Open-Review platform where everyone can view. The reviewer still said he/she cannot take promises. It is odd to us why we would deceive the reviewer in the open-review platform.
> > > >
> > > > Sure you do not have to open-source the code immediately, but I would really appreciate some more information on the experiments because details of e.g. hyperparameter tuning are seriously lacking.
> > > >
> > > > [1]  Counterfactual Multi-Agent Policy Gradients. https://arxiv.org/abs/1705.08926
> > > > [2] Multi-Agent Actor-Critic for Mixed Cooperative-Competitive Environments. https://arxiv.org/abs/1706.02275
> > > > [3] Multi-agent reinforcement learning as a rehearsal for decentralized planning. https://doi.org/10.1016/j.neucom.2016.01.031
> > > > [4] Optimal and Approximate Q-value Functions for Decentralized POMDPs. https://arxiv.org/abs/1111.0062
> > > > [5] A Concise Introduction to Decentralized POMDPs. https://link.springer.com/book/10.1007/978-3-319-28929-8
> > > > [

---

> > > > > ### Author Response · Authors · 2022-11-19
> > > > > **Thanks very much for the response**
> > > > >
> > > > > To begin with, we truly appreciate the reviewer for having a thorough discussion with us and providing some insightful opinions, which we believe is the core spirit of peer review.
> > > > >
> > > > > > CTDE and Dec-POMDP-Com
> > > > >
> > > > > Thank you for the suggestion. We are imprecise and would remove the CTDE claim and we change the Dec-POMDP-Com to multi-agent POMDP.
> > > > >
> > > > > > The purpose of exploring this setting
> > > > >
> > > > > The coordination problem is that ties between equally good actions are broken by all agents randomly [1].
> > > > >
> > > > > Note that even under multi-agent POMDP where agents can get joint observations, coordination problems can still arise. Suppose the centralized critic has learned actions pairs $[a_1,a_2]$ and $[b_1,b_2]$ are equally optimal. Without any prior information, the individual policies $\pi_1$ and $\pi_2$ learned from the centralized critic can break the ties randomly and may choose $a_1$ and $b_2$, respectively.
> > > > >
> > > > > As illustrated, coordination is a general problem in MARL, and not task-specific. It is possible to exist in any fully-cooperative tasks. Note that, in high-dim task, it is unrealistic to analyze the joint optimal action pair. However, Bi-AC [2] claims that the coordination problem is more likely to occur in games with high cooperation levels.
> > > > >
> > > > > > SMAC Issues
> > > > >
> > > > > As mentioned, coordination is a general problem in cooperative tasks and it is more likely to occur in games with high cooperation levels. In other words, coordination problems exist in cooperative games, but it is only a matter of degree.
> > > > >
> > > > > The result of the promising performance in MPE can be that the target conflict issues are the common scenarios.
> > > > >
> > > > > For some maps with lower cooperation levels in SMAC, coordination problems may be rare. That is may explain the performance issues in parts of maps. Note that, for high-dim and complex tasks, the coordination problem cannot be designed on purpose and it is hard to evaluate the cooperation level for the game. In addition, we have to evaluate on the SMAC, otherwise, we will be also questioned for not evaluating on high-dim tasks.
> > > > >
> > > > > In addition, we update two extra maps in 3s_vs_4z and corridor in the appendix, we still claim that the performance gap difference between SeqComm and TarMAC is not marginal.
> > > > >
> > > > > > I was referring to figure 5.
> > > > >
> > > > > Note that the value in the x-axis is 0.x in 10m_vs_11m, while other maps are not. They are in the same order of magnitude.
> > > > >
> > > > > > Hyperparameter
> > > > >
> > > > > We would provide it in the appendix.
> > > > >
> > > > > We really appreciate your comments which help us a lot in improving our paper.
> > > > >
> > > > > [1] Lucian Busoniu, Robert Babuska, and Bart De Schutter. A comprehensive survey of multiagent reinforcement learning.
> > > > >
> > > > > [2] Bi-Level Actor-Critic for Multi-Agent Coordination

---

### Author Response · Authors · 2022-11-15
**We would like to encourage reviewers to respond to our responses.**

We would like to encourage reviewers to respond to our responses. Please let us know whether our responses have addressed your comments and concerns. Also, let us know if you have further comments. We think this would greatly help us to improve the paper.

---

### Decision · Program_Chairs · 2023-01-20

**Decision:**

Reject

**Justification For Why Not Higher Score:**


Weaknesses:
There was considerable confusion amongst the reviewers about the setting and the appropriate baselines.
In particular, it would be great to add fully centralised baselines to the work, e.g. methods that auto-regressively sample the actions for the different agents to get around the large action space.

**Justification For Why Not Lower Score:**

N/A

**Metareview: Summary, Strengths And Weaknesses:**

Summary:
The authors introduce SeqComm, a novel multi-agent learning algorithm that uses train and test-time communication bandwidth between the agents to overcome the limitations of CTDE. The authors test their method in the SMAC benchmakr and compare to popular CTDE methods.

Strengths:
The assumption that agents have some amount of communication available at test time is realistic, which makes this setting realistic.
The experiments overall seem promising.

Weaknesses:
There was considerable confusion amongst the reviewers about the setting and the appropriate baselines.
In particular, it would be great to add fully centralised baselines to the work, e.g. methods that auto-regressively sample the actions for the different agents to get around the large action space. Note that auto-regressive sampling trivially addresses the concerns added by the authors during their rebuttal: "Without any prior information,
the individual policies π1 and π2 learnt from the centralized critic can break the ties randomly and may choose a1 and b2, respectively".

In summary I believe the paper will benefit from another round of reviews at a future venue.